# Conserved regulatory motifs in the juxtamembrane domain and kinase N-lobe revealed through deep mutational scanning of the MET receptor tyrosine kinase domain

Gabriella O Estevam[1,2], Edmond M Linossi[3,4], Christian B Macdonald[1], Carla A Espinoza[2,3,4], Jennifer M Michaud[1], Willow Coyote-Maestas[1,5], Eric A Collisson[6,7], Natalia Jura[3,4,5], James S Fraser[1,5]*

[1]Tetrad Graduate Program, University of California, San Francisco, San Francisco, United States; [2]Cardiovascular Research Institute, University of California, San Francisco, San Francisco, United States; [3]Department of Cellular and Molecular Pharmacology, University of California, San Francisco, San Francisco, United States; [4]Department of Bioengineering and Therapeutic Sciences, University of California, San Francisco, San Francisco, United States; [5]Quantitative Biosciences Institute, University of California, San Francisco, San Francisco, United States; [6]Helen Diller Family Comprehensive Cancer Center, University of California, San Francisco, San Francisco, United States; [7]Department of Medicine/Hematology and Oncology, University of California, San Francisco, San Francisco, United States

**\*For correspondence:**
jfraser@fraserlab.com

**Abstract** MET is a receptor tyrosine kinase (RTK) responsible for initiating signaling pathways involved in development and wound repair. MET activation relies on ligand binding to the extracellular receptor, which prompts dimerization, intracellular phosphorylation, and recruitment of associated signaling proteins. Mutations, which are predominantly observed clinically in the intracellular juxtamembrane and kinase domains, can disrupt typical MET regulatory mechanisms. Understanding how juxtamembrane variants, such as exon 14 skipping (METΔEx14), and rare kinase domain mutations can increase signaling, often leading to cancer, remains a challenge. Here, we perform a parallel deep mutational scan (DMS) of the MET intracellular kinase domain in two fusion protein backgrounds: wild-type and METΔEx14. Our comparative approach has revealed a critical hydrophobic interaction between a juxtamembrane segment and the kinase αC-helix, pointing to potential differences in regulatory mechanisms between MET and other RTKs. Additionally, we have uncovered a β5 motif that acts as a structural pivot for the kinase domain in MET and other TAM family of kinases. We also describe a number of previously unknown activating mutations, aiding the effort to annotate driver, passenger, and drug resistance mutations in the MET kinase domain.

## eLife assessment

This manuscript describes a deep mutational scanning study of the kinase domain of the MET receptor tyrosine kinase. The study yields an **important** catalog of essentially all possible deleterious mutations in this portion of the receptor., with **convincing** evidence. The manuscript will be of interest to researchers working in the field of receptor tyrosine kinases.

**Figure 1.** MET domain boundaries and deep mutational scan (DMS) experimental workflow. (**A**) Domain boundaries of the full-length MET receptor (MET) and TPR-MET fusion (TPR-MET). Extracellular domain (ECD) and intracellular domain (ICD) are distinguished with important phosphorylation sites highlighted in red. Juxtamembrane domain (JM) boundaries are sectioned to annotate the exon 14 coding region and the remainder of the JM (JM2), which includes a αJM-helix (αJM). (**B**) Schematic of the full-length, membrane-associated MET receptor with posited MET ECD dimerization upon hepatocyte growth factor (HGF) binding; schematic of the cytoplasmically expressed, constitutively dimerized TPR-MET construct. The DMS mutagenesis region of the kinase domain (KD) is bordered in red. (**C**) Experimental screen workflow applied to generate and express kinase domain variants prior to selection, beginning with virus generation in Plat-E cells, transduction into Ba/F3 cells at a 0.1 multiplicity of infection (MOI), puromycin selection to enrich for positively infected cells, followed by the IL-3 selection process and time point collection for deep sequencing. (**D**) Post-selection method for analyzing and validating mutation scores based on observed variant frequencies at each time point, measured as a slope which can then be plotted as a distribution.

## Introduction

RTKs are transmembrane proteins that play an essential role in the initiation and regulation of signaling pathways (*Lemmon and Schlessinger, 2010*). Most RTKs are activated upon extracellular ligand binding, promoting a relay of intracellular phosphorylation events that drive signaling (*Lemmon and Schlessinger, 2010*). Mutations that allow RTKs to signal independent of ligand or other typical regulatory mechanisms are commonly identified in cancer (*Duplaquet et al., 2018*; *Saraon et al., 2021*; *Comoglio et al., 2018*). The transition from physiological ligand-dependent to pathological

ligand-independent signaling is exemplified by the RTK, MET (*Figure 1A*). Kinase activity of MET is normally activated by dimerization due to the binding of hepatocyte growth factor (HGF) to the MET extracellular binding domain (*Linossi et al., 2021*). The resultant signaling is crucial for pathways implicated in development and wound repair (*Trusolino et al., 2010*; *Petrini, 2015*; *Kato, 2017*).

In contrast to physiological HGF-regulated activity, MET oncogenic activity arises through a variety of mechanisms such as gene amplification, mutations, gene fusions, and HGF autocrine loops (*Duplaquet et al., 2018*; *Comoglio et al., 2018*; *Saraon et al., 2021*; *Kang et al., 2023*). In MET fusions, replacement of the extracellular domain (ECD) and transmembrane domain by an in-frame translocation generates proteins with constitutive oligomerization of the intracellular domain (ICD), and kinase domain (KD) activity (*Sun et al., 2023*; *Liu et al., 2022*). While MET fusions are rare in patients, they are important tools in studying MET in cellular models (*Park et al., 1986*; *Vigna et al., 1999*; *Pal et al., 2017*). A more common mutation in cancer is METΔEx14, a splicing variant that skips the entire exon 14 coding region, resulting in a shorter ICD missing approximately half of the juxtamembrane domain (JM) upstream of the KD (*Kong-Beltran et al., 2006*; *Ma et al., 2003*; *Frampton et al., 2015*; *Figure 1A*). The METΔEx14 variant maintains the ligand-binding ECD and is oncogenic in part due to a combination of increased ligand sensitivity and reduced degradation due to the loss of a Cbl ubiquitin ligase interaction (*Frampton et al., 2015*; *Abella et al., 2005*; *Peschard et al., 2001*; *Kong-Beltran et al., 2006*). Finally, as in many RTKs, distinct cancer-associated missense mutations are increasingly mapped to the MET kinase domain (*Duplaquet et al., 2018*; *Chiara et al., 2003*). Annotation of the status of these mutations as driver, passenger, or resistance mutations remains a significant challenge for the use of targeted therapies (*Lu et al., 2017*; *Fernandes et al., 2021*).

Here, we use deep mutational scanning (DMS) (*Fowler and Fields, 2014*) to screen a nearly comprehensive set of MET kinase domain mutations. Previous DMS studies have identified potential activating mutations and provided insight into the allosteric regulation of other kinases (*Brenan et al., 2016*; *Ahler et al., 2019*; *Persky et al., 2020*; *Chakraborty et al., 2021*; *Hobbs et al., 2022*). A phenotypic and inhibitor resistance DMS in the Ser/Thr kinase ERK2, reported tumor-associated mutations enriched at recruitment domains, in addition to identifying mutations that confer resistance without direct drug-protein interactions (*Brenan et al., 2016*). Similarly, when compared across Ser/Thr kinases like ERK2 and BRAF, and Tyr kinases like EGFR and ABL1, screens against CDK4/6 highlighted a 'pocket protector' position near the ATP-binding site, and a generalizable allosteric, activating 'keymaster' position within the N-lobe of kinases as sites of drug resistance (*Persky et al., 2020*). DMS of the Tyr kinase, SRC, elucidated a coordinated role between the αF pocket and the SH4 domain to stabilize SRC's closed conformation (*Ahler et al., 2019*). Moreover, mutations to SRC autoinhibitory regions were identified as general resistance hotspots (*Chakraborty et al., 2021*). DMS of an ancestral reconstruction of the Syk-family kinases, AncSZ, revealed mutations that improve bacterial protein expression, in addition to finding commonalities between AncSZ and eukaryotic Syk kinases at regulatory regions like the αC-β4 loop (*Hobbs et al., 2022*). Together, these studies have been critical in illuminating novel features across Tyr and Ser/Thr kinases, which we now build on for the RTK family with MET.

To identify residues that have a direct effect on kinase function, we leveraged the murine Ba/F3 cell line as a selection system. The Ba/F3 cell line has been used as a model to study RTK signaling because it exhibits: (1) undetected expression of endogenous RTKs including Met, (2) addiction to exogenous interleukin-3 (IL-3) for signaling and growth, and (3) dependence on exogenous kinase expression for growth in the absence of IL-3 (*Daley and Baltimore, 1988*; *Warmuth et al., 2007*; *Koga et al., 2022*). IL-3 withdrawal, therefore, serves as a permissive signaling switch that allows for the effective readout of variants that alter kinase-driven proliferation (*Melnick et al., 2006*). Here, we used the TPR-fusion of the MET ICD (TPR-MET) to screen for potential activating and inactivating mutations in the KD. The TPR-MET fusion provides the advantage of studying MET's kinase domain in a cytoplasmic, constitutively oligomerized, active, and HGF-free system (*Figure 1A–B*; *Cooper et al., 1984*; *Park et al., 1986*; *Peschard et al., 2001*; *Rodrigues and Park, 1993*; *Vigna et al., 1999*; *Mak et al., 2007*; *Pal et al., 2017*; *Lu et al., 2017*; *Fujino et al., 2019*). This system also affords enough dynamic range to identify mutations that cause increased proliferation (*Melnick et al., 2006*). We also assessed the impact of exon 14 loss (ΔEx14) on the MET kinase mutational landscape to better understand this oncogenic lesion. Our comprehensive interrogation of the MET kinase domain reveals

novel regulatory regions and acts as a reference for rare activating mutations in both wild-type and ΔEx14 backgrounds.

## Results

### Measurement of MET kinase domain variant activities in a wild-type intracellular domain

To perform a DMS in TPR-MET, we generated a site saturation mutagenesis DNA library of the MET KD (*Figure 1A*). Our library also included the final alpha-helix of the JM region (αJM), which is resolved in most crystal structures, and is just upstream of the KD. Our variant library carried >99% of all possible missense mutations from positions 1059–1345 of human MET, including two internal controls: a WT-synonymous substitution at each position and premature stop codons every eleven positions (24 total). These controls allow us to use deep sequencing to estimate the fitness of WT, null, and deleterious variants. The library was cloned into the TPR-fusion background containing the remaining JM (aa 963–1058), upstream of the αJM and KD, and the complete C-terminal tail (aa 1346–1390) downstream of the KD (*Figure 1A and B*). We transduced the library into Ba/F3 cells using retrovirus (*Figure 1A–C*). Cells were grown in the presence and absence of IL-3 in parallel, and samples were deep sequenced at distinct time points across three biological replicates to identify variant frequencies (*Figure 1C–D*). We then measured variant fitness scores using Enrich2 (*Rubin et al., 2017*) for each selection condition (*Figure 2A*).

As expected, we observed no selection in the IL-3 control condition and a low correlation was observed across replicates (Pearson's $r=0.30$). In this condition, all variants, including premature stop codons, displayed near WT-like fitness (*Figure 2—figure supplement 1A–C*). In contrast, for the condition where IL-3 was withdrawn, we observed evidence of functional selection. There are large differences between the fitness distributions for WT-synonymous, missense, and nonsense mutations. Nonsense mutations are loss-of-function (LOF) (*Figure 2—figure supplement 1B and C*). Few missense mutations are more fit than the average synonymous (WT) variants, which indicates that gain-of-function (GOF) mutations are rare. In addition, fitness estimates for specific mutations across replicates of this treatment are strongly correlated (Pearson's $r=0.96$) (*Figure 2A*, *Figure 2—figure supplement 1B–C*).

### Mutational landscape of the MET kinase domain

MET contains a canonical tyrosine kinase domain, sharing structural and conformational hallmarks with other protein kinases (*Linossi et al., 2021*; *Schiering et al., 2003*; *Wang et al., 2006*). Kinase domains have a conserved hydrophobic core with several motifs that are important for folding, stability, conformational transitions, and catalysis. The kinase domain is structurally composed of two 'lobes,' the N- and C-lobe, which together enclose the catalytic site. The N-lobe is a dynamic unit comprised of five β-sheets and the αC-helix, while the C-lobe is a more rigid unit composed of seven α-helices connected by loops (*Kornev et al., 2006*; *Kornev et al., 2008*). Two hydrophobic 'spines' (the R- and C-spine) assemble across the lobes as the kinase transitions to an active state (*Taylor and Kornev, 2011*; *Haling et al., 2014*; *Hu et al., 2015*). In transitioning from an inactive to the active state, the kinase domain undergoes a conformational rearrangement, becoming catalytically poised through the phosphorylation of A-loop tyrosines 1234 and 1235 (*Longati et al., 1994*), and a stabilized inward C-helix conformation which supports a salt bridge between E1127 and K110 (*Kornev et al., 2006*).

Our DMS results highlight that the conserved regulatory and catalytic motifs in MET are highly sensitive to mutations (*Figure 2A–D*). Catalytic site residues involved in either the chemical step of phosphate transfer or the conformational adaptation to ATP binding, such as K1110, E1127, D1222, and G1224, are intolerant of amino acid substitutions (*Figure 2E*). Residues surrounding the kinase 'hinge,' which are involved in coordinating the adenosine potion of ATP (*Azam et al., 2008*; *Dar and Shokat, 2011*), are more tolerant to mutation (*Figure 2F*). This tolerance is especially prominent in residues that make backbone interactions with ATP. Variants in R-spine residues are enriched in LOF fitness values. This result speaks to the importance of residue identity, not just physicochemical characteristics in the function of MET. R-spine residues F1223 and H1202 are part of the catalytic DFG and HRD motifs and do not tolerate any mutations (*Figure 2G–H*). For R-spine residues M1131 and L1142 positions, only a small number of polar uncharged substitutions show WT-like fitness (*Figure 2G–H*).

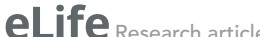

**Figure 2.** Measured effect of MET kinase domain variants across 287 amino acid positions in the context of the full-length juxtamembrane. (**A**) Heatmap of MET kinase domain mutation scores. Wild-type (WT)-synonymous substitutions are outlined in green. (**B**) Active structure of the MET kinase domain (PDB 3R7O), and two representative inactive structures (PDB 2G15, 5HTI) with motif details highlighted. (**C**) Surface representation of average mutational scores mapped on an active structure (PDB 3R7O). Synonymous and nonsense mutations were left out of the averaging and surface representation. Residues at the N- and C-term that were not screened, but modeled in the crystal structure are in white and not considered in the averaging and

*Figure 2 continued on next page*

*Figure 2 continued*

mapping. (**D**) Comparison of surface and core residue scores distributions. A vertical dashed line in both graphs represents the mean score of WT-synonymous mutations. (**E**) Catalytic site and key residues involved in ATP binding and stabilization. Average score of variants mapped onto an active structure (PDB 3R7O). (**F**) Hinge region residues are involved in ATP binding and stabilization. Average score of variants mapped onto an active structure (PDB 3R7O), and overlaid with the ATP molecule of the ATP-bound MET structure (PDB 3DKC). (**G**) Mutation scores and physiochemistry of variants are shown for each residue position of the R- and C- spine of MET. (**H**) R-spine (blue) and C-spine (green) residues are highlighted on an active structure (PDB 3R7O) overlaid with the ATP molecule of the ATP-bound MET structure (PDB 3DKC).

The online version of this article includes the following figure supplement(s) for figure 2:

**Figure supplement 1.** Validation of the MET kinase domain saturation mutagenesis library in IL-3 and IL-3 withdrawal selections.

**Figure supplement 2.** Analysis of RTK R- and C-spine protein sequence conservation.

Surprisingly, the MET C-spine was only moderately sensitive to mutations, with most hydrophobic and polar uncharged amino acid substitutions showing WT-like fitness (*Figure 2G*). In comparing the MET R- and C-spines to other RTK kinase domains through sequence alignments and residue conservation scores, we found the C-spine was highly conserved at positions 1092, 1108, and 1165 but less conserved at positions 1210, 1211, 1212, 1272, and 1276, potentially because these positions co-vary in residue identity across RTKs but pack into the same hydrophobic cluster (*Figure 2G–H*; *Figure 2—figure supplement 2A–B*). Interestingly, despite positions such as 1092 and 1165 showing high conservation, there was still tolerance for several hydrophobic and polar uncharged mutations, further indicating that key catalytic and regulatory motifs display varying sensitivity (*Figure 2G*; *Figure 2—figure supplement 2A–B*). The R-spine and catalytic residues are more highly constrained, whereas C-spine and hinge residues can tolerate a greater number of semi-conservative mutations.

On balance, most solvent-exposed and loop regions were more permissive to mutations (*Figure 2C–D*). However, certain key regulatory elements stand out, with stronger mutational sensitivity than might be expected based on structural features alone. For example, the αC-helix-β4 loop which is important in αC-helix modulation and regulation in many kinases (*Chen et al., 2007*; *Yeung et al., 2020*), showed LOF effects in agreement with the previously described 'ΦxHxNΦΦx' motif (*Yeung et al., 2020*). Similarly, the glycine residues in the 'GxGxxG' motif of the P-loop, which gates ATP entry into the active site, were intolerant to substitutions. Two other relatively immutable sites included Y1234 and Y1235, in the activation loop (A-loop) of MET. The phosphorylation of these residues is required for stabilizing the A-loop in an extended conformation that enables substrate binding and efficient catalysis (*Ferracini et al., 1991*; *Naldini et al., 1991*). Collectively, this deep mutational scan confirms the importance of canonical kinase features and provides a reference point for discovering previously unexplored features of the MET kinase.

## Critical contacts between the αJM and αC helices provide insight into potential juxtamembrane regulation of MET

Given the importance of juxtamembrane (JM) regions in controlling the active-state of many RTKs (*Hubbard, 2004*; *Zhang et al., 2006*; *Jura et al., 2009*; *Cabail et al., 2015*; *Wybenga-Groot et al., 2001*; *Wiesner et al., 2006*) and the prevalence of exon 14 skipping within the JM of MET in cancer (*Lu et al., 2017*), we were interested in how patterns of mutational sensitivity from the DMS could relate the JM to potential MET regulatory mechanisms. The MET JM (aa 956–1075) is predicted to be largely unstructured, but a small region (aa 1059–1070) that folds into an alpha helix (αJM-helix) that packs on top of the αC-helix of the kinase domain, forming a hydrophobic interface (*Figure 3A*). Since we included αJM-helix in the DMS library, we examined the pattern of substitutions for this region and the αC-helix. We observed a hydrophobic preference along both the αJM-helix and αC-helix residues comprising this interface (*Figure 3A*). The adjacent αJM-helix residue 1071 also strongly prefers hydrophobic residues. This residue does not interact with αC-helix, but evidently plays an important role in maintaining interactions with a hydrophobic patch in the N-lobe that includes residues L1076, L1097, and V1158. This result indicates the importance of burying the hydrophobic surfaces of the N-lobe, and αC-helix in particular, for maintaining an active kinase in the TPR-MET background.

Since the αC-helix conformation is a predictor of activity state in kinases (*Ung et al., 2018*; *Modi and Dunbrack, 2019*), and is often modulated by protein-protein interactions or autoregulatory domains (*Huse and Kuriyan, 2002*), we hypothesized that these hydrophobic contacts might form



**Figure 3.** Essential αJM and αC interactions were revealed through variant and structural analysis. (**A**) Ensemble of 93 MET kinase domain crystal structures available in the PDB. All structures, independent of conformation, were locally aligned to JM residues 1059–1070 (all resolved JM and αJM-helix residues in orange), and αC-helix residues 1117–1134 (teal). In solid gray is a representative active structure (PDB 3R7O). Residues involved in the αJM-helix and αC-helix interface. (**B**) Heatmap sections of the αJM-helix and αC-helix from the MET intracellular domain (ICD) screen. (**C**) Distribution of alpha carbon distances for residues in the αJM-helix and αC-helix interface, shown for 63 MET crystal structures in the ensemble with residues modeled for positions 1058, 1062, 1066, 1121, 1125, and 1129. Distances are independent of conformation. (**D**) Global alignment of inactive and active RTK kinase domain structures with resolved JM regions. (**E**) Residue-by-residue, backbone RMSD comparisons of inactive and active structures of MET, AXL, IR, EPHA3, KIT, and RET. (**F**) MuscleWS alignment of human MET and TAM family juxtamembrane helix sequences. (**G**) Crystal structures of MET (PDB 3R7O), RON (PDB 3PLS), and AXL (PDB 5U6B) kinase domains, with αJM-helix (orange) and αC-helix (teal) highlighted. The inactive conformation of AXL shows an αJM-helix and αC-helix hydrophobic interaction similar to MET, but unlike MET, these interactions are slightly pivoted by an αJM-helix turn in its active conformation.

preferentially in active MET kinase. The precedence for this idea comes from structural studies of RTK ICDs in the EGF, PDGF, EPH, and IR families (*Zhang et al., 2006*; *Jura et al., 2009*; *Griffith et al., 2004*; *Wybenga-Groot et al., 2001*; *Wiesner et al., 2006*; *Li et al., 2003*; *Cabail et al., 2015*). For example, in EGFR the JM stabilizes an active, asymmetric head-to-tail dimer (*Zhang et al., 2006*; *Jura et al., 2009*). In IR, the αJM engages with αC-helix to maintain an inactive state, until αJM-αC helix interactions are released and swapped to stabilize an active kinase dimer (*Cabail et al., 2015*). In contrast, in FLT3 the JM packs against the catalytic cleft to stabilize an inactive KD confirmation (*Griffith et al., 2004*). To test if the active-state is linked to JM conformation, we compared the 93 crystal structures of the MET kinase domain with a modeled portion of its JM, as a MET pseudo-ensemble, and performed a local alignment to the αJM-helix and αC-helix (*Figure 3A*). Despite large changes in the relative position of the C-lobe, corresponding to active and inactive structural hallmarks, there was little αJM-helix conformational variability or dependence on the active/inactive state (*Figure 3A–C*). Rather, αJM-helix consistently packed against αC-helix and maintained the hydrophobic interface across all structures (*Figure 3A–C*).

To compare the MET αJM-αC helix to other RTK JM-KD interactions, we compiled a set of human RTK kinase domain crystal structures that contain a modeled αJM-helix, and have both active and inactive structures available in the PDB. Within this set, IR and KIT displayed large conformational variability of the αJM-helix between active and inactive states, but MET stood out again in low conformational variability independent of the kinase conformation (*Figure 3D and E*). Indeed, when we examined JMs and kinase domains in a protein sequence alignment of all RTK, we observed that while αC-helix is conserved across RTKs, only MET (MET, RON), TAM (TYRO3, AXL, MER), and the RYK pseudokinase harbor a αJM-helix with a hydrophobic sequence pattern (*Figure 3F and G*). Together, the mutational sensitivity and structural conservation of MET's αJM-αC helix interface point to a model, that may be shared with RON and TAM family RTKs, where αJM-αC helix contacts are maintained in both the active and inactive state and are important for TPR-MET stability and activity.

## β5 P1153 is a structural pivot for the MET kinase domain N-lobe

Although most regions of high sensitivity from the DMS impinge on well-described aspects of kinase activity, P1153 stood out as a previously unremarked upon position with extremely low tolerance to mutation in our DMS (*Figure 4A*). Only proline was tolerated in this position. P1153 is located in the β4-β5 region, which plays a role in αC-helix coordination and R-spine support (*McClendon et al., 2014*; *Meharena et al., 2013*; *Taylor and Kornev, 2011*). The β4 strand is connected to αC-helix and influences activity through regulatory mechanisms such as the 'molecular break' in RTKs like FGFR, or through cis- and trans- protein interactions (*Chen et al., 2007*; *Yeung et al., 2020*). The β5 strand engages with the αC-helix, the hinge, and harbors the disease-associated 'gatekeeper' position (*Azam et al., 2008*). The immutability phenotype was especially notable because P1153 is not conserved across kinases (*Figure 4—figure supplement 1A*).

To test the importance of this residue in a more physiological membrane-bound context, we expressed WT and P1153L variants of the full-length MET receptor in HEK293 cells. Based on the DMS in Ba/F3 cells, we expected that P1153 mutations would not signal or express poorly in this context. Indeed, relative to the WT MET receptor, P1153L expressed equivalently, but dramatically reduced signals for phosphorylation of the MET A-loop tyrosines, a marker of active MET signaling (*Figure 4B*). To test whether this importance extended to another MET family member, the RON receptor, which also has proline residue at the equivalent position, we tested and observed similar results. Notably, in RON the equivalent proline mutation also significantly compromised receptor expression (*Figure 4—figure supplement 1B*). These experiments provided initial validation that the results from the TPR-MET construct would translate to a full-length MET receptor context, and indicated the family-specific functional importance of proline in this position, of the β4-β5 loop.

Next, we analyzed the structural environment of P1153. It packs in a hydrophobic cluster with the resistance-associated β3-αC helix 'keymaster' position (L1112 in MET) (*Persky et al., 2020*) and F1124 of αC-helix. The hydrophobic packing around P1153 by F1124 and L1112 changes across MET active and inactive structures (*Figure 4C*). As the αC-helix adopts an active 'in' or inactive 'out' conformation, L1112 rotates inward toward the core of the N-lobe to replace F1124 and maintain the hydrophobic environment around P1153 (*Figure 4C*). The maintenance of hydrophobic contacts across these conformational changes led us to re-examine the relative lack of sequence conservation



**Figure 4.** β5 Proline motif is a structural pivot for the MET kinase domain. (**A**) Respective sections of the MET intracellular domain (ICD) heatmap. (**B**) FLAG-IP western blot of the P1153L mutation post 24 hr expression in HEK293 cells. (**C**) Residues of the MET P1153 N-lobe network are displayed in an active (PDB 3R7O) and inactive structure (PDB 2G15). Surface representation of residues involved in the P1153 network. (**D**) Ramachandran plot and

*Figure 4 continued on next page*

*Figure 4 continued*

structural position of P1153 in MET and one representative kinase domain of each receptor tyrosine kinas (RTK) subfamily. (**E**) Structural representation of the RTK Pro shift of the β4–5 loop. One representative RTK kinase domain from each sub-family is locally aligned to β4–5 of MET.

The online version of this article includes the following source data and figure supplement(s) for figure 4:

**Source data 1.** Uncropped FLAG-IP western blot of the P1153L mutant in MET.

**Source data 2.** Raw unedited blots for (*Figure 4*).

**Figure supplement 1.** Receptor tyrosine kinase (RTK) β5-turn site sequence analysis and validation in RON.

**Figure supplement 1—source data 1.** Uncropped FLAG-IP western blot of the P1157L mutant expressed in RON.

**Figure supplement 1—source data 2.** Raw unedited blots for (*Figure 4—figure supplement 1*).

of P1153 (*Figure 4—figure supplement 1A*). We compiled a set of representative kinase domain structures from each RTK family, as well as those that have had DMS studies performed previously, and analyzed the Ramachandran angles of the analogous residue in the β4-β5 loop. Although most kinases have a non-proline amino acid at that position, the analogous residues cluster around or inside the proline-permissive contour of Ramachandran space (*Figure 4D*). Interestingly, while no other RTK family aside from MET and TAM family kinases have a proline at the β4-β5 loop position, most do have a proline exactly one position upstream (*Figure 4E*). This shift occurs both in sequence and in structure, where the upstream proline does not participate in the hydrophobic pivot observed for P1153 in MET (*Figure 4D–E*). This analysis suggests that the MET and TAM families represent a subset of RTKs that have evolved the surrounding sequence to accommodate only proline in this structurally restrained region of the β4-β5 loop, explaining the immutability in our DMS experiment.

## Mutational landscape of the MET kinase domain in the absence of the exon 14 coding region

Exon 14 skipping, which maintains the ECD and transmembrane region while truncating the ICD at the JM, is one of the most common driver mutations observed in MET (*Fernandes et al., 2021*; *Lu et al., 2017*). The oncogenic effect is thought to, at least partially, result from the removal of the docking site for Cbl, a ubiquitin ligase responsible for MET lysosomal degradation (*Mak et al., 2007*; *Peschard et al., 2001*; *Petrelli et al., 2002*; *Abella et al., 2005*; *Kong-Beltran et al., 2006*). Therefore, to understand the the shift in the kinase domain mutational landscape under varying juxtamembrane contexts, particularly the oncogenic exon 14 skipped background, we performed a parallel DMS of the exon 14-skipped ICD (TPR-METΔEx14) and applied the same library benchmarking used for the wild-type construct (TPR-MET) (*Figure 5C*; *Figure 5—figure supplement 1*).

Based on previous findings showing a TPR-METΔEx14 growth advantage over the MET receptor and METΔEx14 receptor, and TPR-MET in anchorage-independent assays of AALE cells (*Lu et al., 2017*) and fibroblasts (*Vigna et al., 1999*), we tested whether there was also a growth advantage in Ba/F3 cells (*Figure 5A*). Counter to expectations, we did not observe a discernible growth differential for TPR-METΔEx14 relative to TPR-MET (*Figure 5B*; *Figure 5—figure supplement 2A*). It is important to note that exon 14 skipping normally occurs in the context of the full-length protein, not in fusions such as TPR-MET. Therefore, the lack of a growth rate difference may be due to the need for Cbl regulation to occur at the plasma membrane (*Mak et al., 2007*) or due to other aspects of the Ba/F3 system. Although the lack of a growth differential means an exon 14-specific mechanism cannot be fully addressed in this experimental format, KD mutational responses are still representative and comparable.

In analyzing the mutational landscape of the TPR-METΔEx14, we found that regions implicated in the TPR-MET DMS were similarly sensitive (*Figure 5C and D*; *Figure 5—figure supplement 2B*). For example, the P1153 β4-β5 loop site was also intolerant to mutations in TPR-METΔEx14 (*Figure 5—figure supplement 2C*). Similarly, the R- and C-spine were sensitive to physicochemical changes outside of hydrophobic and polar uncharged substitutions (*Figure 5—figure supplement 2D*).

With the caveats of membrane context and growth differentials in mind, to address exon 14 and juxtamembrane-driven differences between these two mutational landscapes, we calculated the absolute score difference between TPR-METΔEx14 and TPR-MET mutations (|METΔEx14 - MET|) and plotted the |ΔScore| in a heatmap (*Figure 5—figure supplement 3A*). Overall, the two landscapes were largely similar with differences emerging mostly for specific mutations. To identify and structurally map



**Figure 5.** Comparative measurement of MET kinase domain variants across 287 amino acid positions in the absence (TPR-METΔEx14) and presence of exon 14 (TPR-MET). (**A**) Domain boundaries and schematics of the TPR-MET ICD and TPR-METΔEx14 ICD constructs. (**B**) Proliferation assay of parental TPR-MET, TPR-METΔEx14, and MSCV empty vectors expressed in Ba/F3 under IL-3 withdrawal and IL-3 conditions. Cell viability was normalized to day 0 (n = 3). (**C**) Heatmap of METΔEx14 kinase domain variant scores. Wild-type (WT)-synonymous substitutions are outlined in green. (**D**) Scatter plot of TPR-METΔEx14 versus TPR-MET scores for each variant with distributions displayed on the margins. Dashed lines represent the WT synonymous average

*Figure 5 continued on next page*

*Figure 5 continued*

score for METΔEx14 versus MET. (**E**) Schematic of the kinase domain (PDB 2G15, 3R7O) in a full-length receptor and TPR-METΔEx14 context. (**F**) Western blot of endogenous MET KO HeLa cells transiently transfected with L1062D and S1122Q mutants in the MET and METΔEx14 receptor, with and without HGF stimulation (50 ng/ml, 15 min stimulation, 37 °C).

The online version of this article includes the following source data and figure supplement(s) for figure 5:

**Source data 1.** Uncropped western blots and exposures for L1062D and S1122Q mutations.

**Source data 2.** Raw unedited blots for (*Figure 5*).

**Figure supplement 1.** Validation of the METΔEx14 saturation mutagenesis library in IL-3 and IL-3 withdrawal selections.

**Figure supplement 2.** Comparative analysis of the TPR-METΔEx14 and TPR-MET mutational landscapes.

**Figure supplement 3.** Landscape of fitness and gain-of-function mutational differences between METΔEx14 and MET kinase domain libraries.

specific mutations and structural regions with the most divergent responses, positions with a |ΔScore| that met our statistical criteria were filtered and structurally mapped (*Figure 5—figure supplement 3B–D*). Through this analysis, the largest secondary structural difference continued to emerge as the αJM-helix, where MET is more sensitive to helix-breaking mutations such as proline than METΔEx14 (*Figure 3*; *Figure 5—figure supplement 2F*; *Figure 5—figure supplement 3B–D*). Of all αJM-helix positions, L1062 displayed the greatest difference in sensitivity between these two datasets, with the introduction of negative charge resulting in the largest loss-of-function effect for the TPR-MET kinase domain, but having a null effect in the TPR-METΔEx14 kinase domain. Other regions with strong differences include the αG and APE motifs (*Figure 5—figure supplement 3D*).

To test whether this αJM-helix and αC-helix sensitivity difference translated to the full-length receptor, we introduced L1062D, a mutation at the αJM-αC-helix interface, and S1122Q, an αC-helix surface mutation, into the MET and METΔEx14 receptor backgrounds in HeLa cells lacking endogenous MET (*Figure 5E and F*). Consistent with the results from the DMS screen, in unstimulated cells, we observed that a marker of MET activity, A-loop phosphorylation, was dramatically reduced in the MET receptor for L1062D, but less so in METΔEx14, relative to WT controls (*Figure 5F*). Furthermore, upon HGF treatment, L1062D in the METΔEx14 background exhibited a high degree of activity, but L1062D MET did not. An additional marker of active MET signaling, phosphorylation on p44/42 MAPK (ERK), was similarly responsive (*Figure 5F*). In contrast, S1122Q was expected to have a gain-of-function effect in the MET receptor and a loss-of-function effect in the METΔEx14 receptor. For this mutation we observed no difference relative to WT in A-loop phosphorylation at baseline or upon HGF stimulation, and unexpectedly a higher pERK signal in METΔEx14 than MET, indicating that probing multiple phosphorylation sites and MET-associate pathways may be important to understand in what way mutations affect MET phosphorylation and proliferation in the receptor (*Figure 5F*).

These results suggest that the correlation between DMS fitness in TPR-MET and acute A-loop phosphorylation in the MET receptor is not absolute. Since KD activity at a single time point was the only parameter we explored with the membrane-associated receptor, other aspects such as sustained signaling, recruitment of specific signaling adapters, or changes in MET regulation may be more consistent with the proliferative readout of the DMS. In addition, the inconsistency in the S1122Q result may be a specific feature of the dimerized, cytoplasmic TPR-MET signaling that is sensitive to interactions with the portion of the JM deleted by exon 14. Collectively, these results highlight that the αJM-αC helix interface is sensitive to mutation and the presence or absence of exon 14 can alter this sensitivity. While the experimental parameters of this screen may limit how translatable some mutations are in a membrane-associated receptor or in non-proliferative conditions, the consistency of most residues between backgrounds provides increased confidence in the atlas of mutational effects on MET activity.

## Analysis of cancer-associated and resistance mutations in MET and METΔEx14

To assess the ability of the DMS to classify driver, passenger, and resistance mutations, we first gathered all MET kinase domain mutations reported from clinical observations in cBioPortal (*Cerami et al., 2012*; *Gao et al., 2013*; *Figure 6A*). Next, we compared GOF mutations between the TPR-MET and TPR-METΔEx14 libraries by statistically filtering scores within each library and comparing scores within IL-3 and IL-3 withdrawal conditions. We structurally mapped scores that had low propagated error and

**Figure 6.** Mutations with greater proliferative effects than cancer-associated mutations, and differential sensitivities between MET and METΔEx14 identified. (**A**) Lollipop diagram of MET kinase domain mutations and frequencies annotated in cBioPortal. (**B**) Distributions of clinically-associated mutations (green, right y-axis scale) overlaid with all missense mutations (gray, left y-axis scale). (**C–D**) Distributions of categorized cancer-associated mutations. Hamming distance distributions of clinical, validated MET cancer mutations and clinically unobserved, gain-of-function (GOF) mutations detected in the screen for both intracellular domain (ICD) backgrounds. (**E**) Cancer-associated mutations mapped onto a MET kinase domain structure, colored according to MET and METΔEx14 backgrounds, with Hamming distance represented by the ribbon thickness at each position (PDB 3R7O). (**F**) Reported resistance mutation distributions (teal, right y-axis scale) for MET and METΔEx14, overlaid with their respective missense distributions (gray, left y-axis scale). (**G**) Inhibitor resistance mutation positions shown on an active MET kinase domain structure in teal (PDB 3R7O).

*Figure 6 continued on next page*

*Figure 6 continued*

The online version of this article includes the following figure supplement(s) for figure 6:

**Figure supplement 1.** Statistical analysis and classification of gain-of-function mutations across libraries.

**Figure supplement 2.** Identification and analysis of clinically associated mutations across MET kinase domain libraries.

**Figure supplement 3.** Inhibitor-protein interactions for Y1230 and crizotinib.

were greater than 2 or 2.5 standard deviations from the mean of the WT-synonymous score for each library - the same filtering criteria used throughout this study - to assign GOF status (*Figure 6—figure supplement 1A–E*).

Relative to the distribution of all missense mutations, the distribution of clinically observed mutations is shifted to higher fitness values for both TPR-MET and TPR-METΔEx14 screens (*Figure 6B*; *Figure 6—figure supplement 2A–D*). Most of these mutations have near WT-fitness levels, with a small number having GOF fitness effects (*Figure 6B*). Next, we further subdivided our observations based on annotations in cBioPortal as either 'clinical, validated' or 'clinical, not validated.' Notably, mutations in the 'clinical, not validated' category were outliers with lower fitness values, indicating that these are likely passenger mutations (*Figure 6C and D*; *Figure 6—figure supplement 2A–D*). These results validate that the DMS recapitulates known oncogenic MET kinase variants and suggest that the screen can be used to help classify driver vs. passenger mutations.

Within our gain-of-function classifications, while we pick up on several mutations at positions that have been clinically detected and experimentally validated in previous studies in both libraries (*Figure 6—figure supplement 2B–D*), we also found missense mutations that are GOF in the DMS that have not been reported clinically, potentially indicative of novel activating mutations. We classify these mutations as 'not clinically observed, GOF.' We hypothesized that these variants are more difficult to observe clinically because of the constraints of the genetic code. To test this idea, we calculated the Hamming distance between the WT MET codon and mutant codon for each position within our dataset (*Figure 6C–E*). We found that most "clinical, validated" mutations had Hamming distances of 1–2 nucleotide substitutions from WT (*Figure 6C and D*). However, the most common Hamming distances for 'not clinically observed, GOF'' codons shifted 2–3 changes away from WT (*Figure 6C and D*). These GOF mutations are dispersed throughout the structure of MET kinase and have distinct patterns depending on the presence or absence of exon 14 (*Figure 6E*). These results suggest that the DMS can identify GOF mutations that require a larger genetic 'leap' than what is observed in natural populations. Furthermore, the relative paucity of 'not clinically observed, GOF'' mutations at a Hamming distance of 1 suggests that clinical observations have nearly identified all possible activating MET mutations that require only a single nucleotide change. Collectively, this suggests that our atlas will be of particular use for deciding on driver status for rare mutations that require multiple nucleotide changes.

Finally, we assessed the distribution of fitness effects for clinically observed resistance mutations (*Duplaquet et al., 2018*; *Fernandes et al., 2021*; *Saraon et al., 2021*; *Lu et al., 2017*; *Fujino et al., 2019*). Resistance mutations, which are clustered around the active site, are also enriched towards higher fitness values than the background missense mutation distribution (*Figure 6F and G*). This result suggests that most of these mutations can pre-exist in the population even in the absence of selective pressure from an inhibitor. Mutations labeled as both GOF from our statistical filtering, and resistant from literature reports, may indicate an effect on the activity equilibrium of kinase, whereas resistance mutations that are labeled LOF in this screen, may only become beneficial in the presence of an inhibitor. For instance, Y1230C is a recurrent resistance mutation (*Bardelli et al., 1998*) that interrupts Pi interactions that stabilize inhibitors at the active site, but in the absence of an inhibitor is unfavorable in METΔEx14 and only mildly unfavorable in MET (*Figure 6F*; *Figure 6—figure supplement 3*). These results indicate that DMS has the potential to interpret the effects of resistance mutations, an area of active concern for patients being treated with MET inhibitors in the clinic.

## Discussion

Our parallel DMS of the MET kinase domain has revealed how mutations affect the allosteric regulation of MET in two clinically relevant juxtamembrane backgrounds. Auto-regulation via juxtamembrane

segments is a common feature of RTKs like the EGF, PDGF, EPH, and IR families (*Hubbard, 2004*; *Zhang et al., 2006*; *Jura et al., 2009*). We propose that MET is similarly regulated based on the distinct sensitivity of the interface of αJM and αC helices to mutations between TPR-MET and TPR-METΔEx14. Given that exon 14 is directly upstream of this region, it is possible that the making and breaking of contacts between the αJM and αC helices during kinase activation observed in other families has a distinct structural analog in MET. Rather than variable contacts between the JM and αC-helix, as observed in other families, we hypothesize that the exon 14 region of the JM has active-state dependent contacts with the αJM-αC unit. In this model, αJM-αC helix moves in unison between active and inactive conformations, with the rest of the JM making contacts to αJM that regulate the activity state. When exon 14 is skipped, these contacts are absent, tilting the equilibrium towards an active kinase conformation. This model could augment known mechanisms of METΔEx14 activity, most notably a lack of Cbl-mediated downregulation, to enhance the oncogenic potential of this variant.

Conclusively addressing the question of how exon 14 skipping activates MET is difficult in the TPR-fusion system that we have employed here. While soluble, cytosolic oligomerization occurs for MET fusion proteins observed in some tumors, using the TPR-fusion may not accurately model all of the changes found in other types of MET lesions, such as METΔEx14. Therefore, this strategy may under-estimate the effects of regulatory mechanisms related to membrane engagement or ICD oligomer-ization. Nonetheless, the experimental parameters of this study based on cell proliferation allowed us to understand the mutational sensitivities of the kinase domain in a cytoplasmic and constitutively active environment.

Analyzing the mutational landscape with the TPR-MET fusion approach employed here has led to multiple new insights, some of which were also validated in the full-length MET receptor. For example, one of the most unexpected regions of mutational sensitivity was P1153 in the β4-β5 loop. Based on the observed proline sequence shift, β4-β5 loop conformational restraints, and conserved hydrophobic network around the β4-β5 loop position, we suggest that this proline residue is an evolu-tionary dead-end. In ancestral kinases, a proline could be adopted without a significant cost at the corresponding β4-β5 loop position due to this residue being positioned in proline-permissive Ramach-andran space. After the drift of the surrounding hydrophobic residues, the sequence for these resi-dues has adapted to the unique structural properties of proline, rendering it impossible to substitute for an alternative amino acid. Interestingly, evolutionarily distant kinases that do not have proline at this position exhibit inhibitor resistance mutations at this site (*Brenan et al., 2016*; *Persky et al., 2020*; *Lee and Shah, 2017*). In some distant kinases such as PDK1, this β4-β5, and αC-helix region is part of an allosteric binding site for inhibitors (the PIF pocket) (*Rettenmaier et al., 2014*). Given that the proline pivot region is largely intolerant of mutations, this site could potentially be targeted for allosteric inhibitors to avoid the development of resistance mutations observed in the clinic for many other small molecule MET inhibitors.

In summary, our parallel DMS of MET and METΔEx14 has built an atlas of variant effects. Moreover, we identified a small number of unique sensitivities in each background, which provides hypotheses about the mechanism of exon 14 skipping in cancer. We also observed a number of strong GOF variants that have not been observed in the clinic. Strikingly, these variants are enriched in 2 and 3 nucleotide changes, suggesting that our DMS will be especially useful in classifying rare driver muta-tions and that the clinical population has essentially sampled most of the single nucleotide changes. These results comprise a valuable resource for classifying driver, passenger, and resistance mutations for MET and other RTKs.

# Materials and methods
## Mammalian cell culturing
Ba/F3 cells were purchased from DSMZ (Cat. ACC 300) and maintained in 90% RPMI (Gibco), 10% HI-FBS (Gibco), 1% penicillin/streptomycin (Gibco), and 10 ng/ml IL-3 (Fisher), and incubated at 37 °C with 5% $CO_2$. Ba/F3 cells were passaged at or below 1.0E6 cells/ml in order to avoid acquired IL-3 resistance, and regularly checked for IL-3 addiction by performing 3 x PBS (Gibco) washes and outgrowth in the absence of IL-3 to confirm cell death in the parental, empty cell line.

Plat-E cells stably expressing retroviral envelope and packaging plasmids were originally gifted by Dr. Wendell Lim. Plat-E cells were maintained in 90% DMEM +HEPES (Gibco), 10% HI-FBS (Gibco), 1% penicillin/streptomycin (Gibco), 10 ug/ml blasticidin, 1 ug/ml puromycin, and incubated at 37 °C with 5% CO2. Plat-E cells were maintained under blasticidin and puromycin antibiotic pressure unless being transfected.

HEK293 cells were maintained in DMEM (Gibco) supplemented with 10% FBS (Gibco) and 1% penicillin/streptomycin (Gibco) at 37 °C in 5% CO2.

Human MET knockout HeLa cells were purchased from Abcam (Cat. ab265961) and maintained in 90% DMEM + HEPES (Gibco), 10% HI-FBS (Gibco), 1% penicillin/streptomycin (Gibco), and incubated at 37 °C with 5% CO2.

Cells were routinely checked for mycoplasma (Lonza).

## Cloning and retroviral vectors used

Both pUC19 (Cat. 50005) and MSCV (Cat. 68469) were ordered from Addgene. To ensure unique cut sites within the vectors for introduction and shuttling of the variant library, a new multiple cloning site was introduced into each plasmid. Wild-type TPR-MET-IRES-mCherry and TPR-METΔEx14-IRES-eGFP genes were cloned into pUC19 as the parental constructs for library generation and all site-directed mutagenesis. The full-length MET, METΔEx14, and RON receptor cDNAs were subcloned into pcDNA3.1 vector by Gibson assembly and incorporated a C-terminal single Flag tag sequence. All mutations were introduced by quick change mutagenesis.

## MET kinase domain variant library generation and cloning

All coning was performed and handled in parallel for TPR-MET and TPR-METΔEx14 libraries unless otherwise stated.

The MET kinase domain variant library was designed to span amino acid positions 1059–1345, which contained the full kinase domain (aa 1071–1345) and a portion of the juxtamembrane (aa 1059–1070). The library was synthesized by Twist Bioscience with one mammalian high-usage codon per amino acid to prevent over-representation of specific residues. The library was received in 96-well arrays of lyophilized DNA at 50 ng per well, where each well contained all variants (missense and WT-synonymous) per position of the kinase domain. The lyophilized library was resuspended in 100 uL of 1 X TE buffer, and 5 ng of DNA from each well was amplified with low cycle PCR to increase the starting material using the following the NEB Q5 High-Fidelity recipe per well: 10 µL 5 X Q5 buffer, 5 ng template DNA, 2.5 µL 10 µM forward primer, 2.5 µL 10 µM reverse primer, 1 µL 10 mM dNTPs (2.5 µM each), 0.5 µL Q5 Polymerase, nuclease free water to a final volume of 50 µL. The following thermocycler parameters were then applied: initial denaturation at 98 °C for 30 s, followed by 10 x cycles of denaturation at 98 °C for 10 s, annealing at 62 °C for 30 s, extension at 72 °C for 1 min, and a final extension at 72 °C for 5 min. A 1% agarose, 1 X TBE diagnostic gel was run with 2 µL of each sample to confirm amplification of all positions, then the samples were PCR cleaned using the Zymo 96-well DNA clean and concentrate kit, eluted in 10 µL nuclease-free water, pooled together in a low DNA-bind tube, and then DNA cleaned (Zymo) once more to further concentrate the pooled library.

The kinase domain variant library was digested with PstI-HF (NEB) and NdeI-HF (NEB) and cleaned up with the Zymo DNA clean and concentrate kit. Next, the two cloning vectors (pUC19_kozak-TPR-METΔEx14-IRES-eGFP and pUC19_kozak-TPR-MET-IRES-mCherry), were digested with PstI-HF(NEB) and NdeI-HF (NEB), phosphatase treated with rSAP (NEB), gel extracted to isolate the backbone, and DNA cleaned (Zymo). The variant library was ligated into each vector with a 1:3 (insert: vector) T4 ligation at 16 °C overnight (NEB). Ligations were DNA cleaned (Zymo), eluted in 10 uL of nuclease-free water, and electroporated into 50 µL MegaX 10 beta cells (Invitrogen). Transformations were then recovered in 1 mL of SOC for 1 hr at 37 °C. Post recovery, 10 µL cells were collected, serial diluted, and plated at varying dilutions (1:100, 1:1 k, 1:10 k, 1:100 k, 1:1 M) to evaluate transformation efficiencies. The remainder of the transformation was then propagated in 50 mL LB and Carbeniacillin at 37 °C to an OD of 0.5, and then midi-prepped (Zymo).

Amino acid variants were successfully synthesized by Twist Biosciences for all positions with the exception of 1194 and 1278. In addition, premature stop codons were not included in the synthesized Twist library. To include these missing positions and early stop control, we generated a 'fill-in' library. For positions 1194 and 1278, a forward primer for each amino acid mutation and a single

reverse primer was designed for inverse PCR. An early stop codon 'fill-in' library was also generated to introduce one stop codon every 33 bases, evenly spaced throughout the gene. This resulted in one stop codon every 11 positions, or 24 total premature stops. Again, a single forward and reverse primer was designed for each stop codon mutation using inverse PCR. Mutations were introduced into wild-type pUC19_kozak-TPR-METΔEx14-IRES-eGFP and pUC19_kozak-TPR-MET-IRES-mCherry with the following NEB Q5 High-Fidelity conditions per reaction: 10 µL 5 X Q5 buffer, 5 ng template DNA, 2.5 µL 10 µM forward primer, 2.5 µL 10 µM reverse primer, 1 µL 10 mM dNTPs (2.5 µM each), 0.5 µL Q5 Polymerase, nuclease-free water to a final volume of 50 µL. The following thermocycler parameters were then applied: initial denaturation at 98 °C for 30 s, 10 x cycles of denaturation at 98 °C for 10 s, annealing at 62 °C for 30 s, extension at 72 °C for 4.4 min, and a final extension at 72 °C for 10 min. A 1% agarose diagnostic gel was run with 2 µL of each reaction to conform amplification. Then all PCR samples were pooled, DNA cleaned (Zymo), and eluted in 50 µL nuclease-free water, DPN1 was digested to remove the template (NEB), DNA was cleaned again and eluted in 12 µL nuclease-free water, PNK treated (NEB), and blunt-end ligated at 16 °C overnight with T4 ligase. Ligations were DNA cleaned the next morning (Zymo) and electroporated into MegaX 10 beta cells (Invitrogen). Transformations were recovered in 1 mL of SOC for 1 hr at 37 °C, plated at varying dilutions to estimate transformation efficiencies, propagated in 50 mL LB and Carbyinacillin at 37 °C to an OD of 0.5, and then midi-prepped (Zymo). The Twist-synthesized library was then pooled together with the 1278, 1194, and premature stop 'fill-in' libraries at equimolar concentrations to a total of 1 ug of DNA.

7 µg of each pooled library was restriction digested with MluI-HF (NEB) and MfeI-HF (NEB) to cut out variant libraries in the kozak-TPR-MET-IRES-mCherry and kozak-TPR-METΔEx14-IRES-eGFP backgrounds. Digests were gel extracted from pUC19 and DNA cleaned (Zymo). The empty, Puromycin resistant, retroviral expression vector MSCV (addgene) was also cut MluI-HF (NEB) and MfeI-HF (NEB), phosphatase treated with rSAP (NEB), and DNA cleaned. Isolated libraries were then ligated 1:1 (insert to vector) into the MSCV retroviral vector at 16 °C overnight with T4 ligase. Ligations were then DNA cleaned (Zymo) and electroporated into ElectroMAX Stbl4 Competent Cells (Thermo Fisher). Transformations were recovered in 1 mL of SOC for 1 hr at 37 °C and after recovery, 10 µL was serially diluted and plated to estimate transformation efficiencies, while the remainder was plated on bioassay dishes. Colonies were scraped from the bioassay dishes and midi-prepped for transfections (Zymo).

## Variant library introduction into Ba/F3

The MSCV_kozak-TPR-MET-IRES-mCherry and MSCV_kozak-TPR-METΔEx14-IRES-eGFP variant libraries were transfected into Plat-E cells for retroviral packaging using Lipofectamine3000 (Invitrogen). Two T-175 flasks of Plat-E cells were prepared for each library in the absence of blasticidin and puromycin 24 hr prior to transfection such that they would be at 70–80% confluency at the time of transfection. On the day of transfection, each flask of Plat-E cells was gently washed with PBS to remove the culturing media, and replaced with 35 mL Opti-MEM. For the transfection, Opti-MEM was brought to room temperature and two pairs of DNA Lo-bind 5 mL tubes were prepared following the manufacturer's instructions for Lipofectamine3000 scaled to a T-175 format. A total of 46 µg DNA was used to transfect the libraries and package the virus in parallel: MSCV_TPR-MET-IRES-eGFP, MSCV_TPR-METΔEx14-IRES-mCherry. Each flask was incubated with the transfection reagents for 5 hr at 37 °C, 5% CO2; the transfection media was then replaced with 50 mL OptiMEM, 5% FBS, 1 x GlutaMax, and 2% Sodium Pyruvate (Gibco) for viral packaging. After 48 hr post-transfection, the viral supernatant was harvested, passed through a 0.45 µm filter to remove cell debris, then precipitated overnight with 1:4 Retro-X concentrator (TakaraBio) at 4 °C, then pelleted at 380xg for 45 min at 4 C, and resuspended in 5 mL of sterile, cold PBS and stored at 4 °C in 1 mL aliquots until transduced into Ba/F3 cells.

The concentrated virus was titered in Ba/F3 cells in a 6-well plate format. Cells were seeded at 1.0E5 cells/ml with 10 ng/ml IL-3 and 8 µg/ml polybrene (Sigma-Aldrich). Virus was added to wells at 0, 10 x, 20 x, and 40 x dilutions to determine the proper volume for a transduction MOI of 0.1–0.3. Cells were spinfected at 250xg for 60 min at room temperature, then incubated for 48 hr. The viral titer was calculated from the percent of fluorescent cells and viral dilution.

For the DMS viral transduction, 6 million cells were spinfected at an MOI of 0.1, in triplicate for a total of three biological replicates for each library, and incubated post spinfection in a 15 cm dish with

Cancer Biology | Structural Biology and Molecular Biophysics

30 mL Ba/F3 media and 10 ng/L IL-3 for 48 hr. Infected cells were then selected with 1 μg/ml puromycin for a total of 4 days with fluorescence and cell counts tracked each day.

## DMS time point selection and sample preparation

All screening conditions were performed and handled in parallel, on the same days for TPR-MET and TPR-METΔEx14 libraries across all independent conditions and biological replicates (library, biological replicate, time point, IL-3 condition).

After puromycin selection, all three biological replicates for both libraries, TPR-MET and TPR-METΔEx14, were washed free of puromycin and IL-3 with 3 x PBS washes. A total of 6 million cells from each replicate was harvested and pelleted at 250 × g to serve as the 'time point 0' pre-selection sample (T0).

To begin the selection of each replicate for each library, two sets of 15 cm dishes were prepared with 2.0E5 cells/ml in 30 mL 90% RPMI, 10% HI-FBS, and 1% penicillin/streptomycin. One plate was kept free of IL-3 as the experimental IL-3 withdrawal condition, while the other plate was supplemented with 10 ng/mL IL-3 to provide the control condition. Three-time points post T0 were collected for each library replicate and condition for a total of four-time points (T0, T1, T2, T3). Time points were harvested every 48 hr across 7 days; 6 million cells were harvested for each condition and pelleted at 250 × g for 5 min; 2.0E5 cells/ml were split at every time point and maintained either in IL-3 or IL-3 withdrawal conditions.

The gDNA of each time point sample was isolated with the TakaraBio NucleoSpin Blood QuickPure kit the same day the cells were harvested. gDNA was eluted in a 50 μl elution buffer using the high concentration and high yield elution manufacturer's protocol. Immediately after gDNA was isolated, 5 μg of gDNA was used for PCR amplification of the target MET KD gene to achieve the proper variant coverage. A 150 μl PCR master mix was prepared for each sample using the TakaraBio PrimeStar GXL system according to the following recipe: 30 μl 5 X PrimeStar GXL buffer, 4.5 μl 10 μM forward primer (0.3 uM final), 4.5 μl 10 μM reverse primer (0.3 uM final), 5 μg gDNA, 12 μl 10 mM dNTPs (2.5 mM each NTP), 6 μl GXL polymerase, nuclease-free water to a final reaction volume of 150 uL. The PCR master mix was split into three PCR tubes with 50 μl volumes and amplified with the following thermocycler parameters: initial denaturation at 98 °C for 30 s, followed by 24 x cycles of denaturation at 98 °C for 10 s, annealing at 60 °C for 15 s, extension at 68 °C for 14 s, and a final extension at 68 °C for 1 min.

## Library preparation and deep sequencing

After all time points were selected, harvested, and PCR amplified for both TPR-MET and TPR-METΔEx14 libraries, the target gene amplicon was isolated from gDNA by gel purification. The entire 150 μl PCR reaction for each sample was mixed with 1 X NEB Purple Loading Dye (6 X stock) and run on a 0.8% agarose, 1 X TBE gel, at 100 mA until there was clear ladder and band separation. The target amplicons were gel excised and purified with the Zymo Gel DNA Recovery kit. To remove excess agarose contamination, each sample was then further cleaned using the Zymo DNA Clean and Concentrator-5 kit. Amplicon DNA concentrations were then determined by Qubit dsDNA HS assay (Invitrogen).

TPR-MET and TPR-METΔEx14 libraries were then prepared for deep sequencing using the Nextera XT DNA Library Prep kit in a 96-well plate format (Illumina). Manufacturer's instructions were followed for each step: tagmentation, indexing and amplification, and clean-up. Libraries were indexed using the IDT for Nextera Unique Dual Indexes Set C (Illumina). In this approach, the 861 bp amplicon (1190 bp total amplicon length with sequence flanking the target at the 5' and 3' ends for buffer), for each library is randomly cleaved by a transposome into ~300 bp fragments following a Poisson distribution, which allows for direct sequencing. Indexed libraries were quantified using the Agilent TapeStation with HS D5000 screen tape and reagents (Agilent). DNA concentrations were further confirmed with a Qubit dsDNA HS assay. All samples across both TPR-MET and TPR-METΔEx14 libraries (library, biological replicate, time point, IL-3 condition) were manually normalized to 10 nM and pooled. The libraries were then paired-end sequenced (SP300) on two lanes of a NovaSeq6000.

## MET kinase domain variant analysis and scoring

Demultiplexed paired-end reads were received from the sequencing core and processed further using a snakemake-based pipeline previously developed (*Macdonald et al., 2023*; *Mölder et al., 2021*).

Initial QC was performed via FastQC (**Andrews, 2010**), and continued via aggregation of intermediate output statistics with MultiQC (**Ewels et al., 2016**). First, any remaining adapter sequences or contaminant sequences were removed with BBDuk. Next, overlap-based error-correction was employed with BBMerge, before being mapped to the reference sequence with BBMap (**Bushnell, 2014**). Variant counts from each mapped BAM file were then made with the GATK AnalyzeSaturationMutagenesis tool (**Van der and O'Connor, 2020**). The output of this tool was processed using a script to remove variants that were not in our initial library design and to prepare output in a format for further processing with Enrich2 using weighted-least squares with wild-type normalization (**Rubin et al., 2017**). It was noted that Enrich2 produces unexpected scores when some variants are unobserved across replicates or go to zero over a single time course: to avoid this, our script also detects this and removes them in advance.

## MET and METΔEx14 mutational analysis

Raw Enrich2 scores were used for all comparative mutation 'score' measurements. gain-of-function and loss-of-function missense mutations were classified and calculated as ±2 SD from the mean score of the WT-synonymous distributions for MET and METΔEx14. For comparative analysis, propagation of error was calculated from the delta score (Δscore) and delta standard error (ΔSE) of each variant for MET and METΔEx14, and only variants with a standard error difference lower than the score difference were used.

## Validation of variants in the MET and METΔEx14 receptor by western blot

HEK293 cells were transiently transfected with Lipofectamine3000 (Invitrogen) according to the manufacturer's protocols. Cells were harvested 24 hr post-transfection, lysed in buffer (50 mM Tris-HCl, pH 7.5, 150 mM NaCl, 2 mM EDTA, and 1 % w/v Triton X-100 supplemented with protease inhibitor tablets (Roche), 1 mM sodium fluoride and 1 mM sodium vanadate). Clarified lysates were incubated with G1 affinity resin (Genscript) overnight at 4 °C. Resin was washed with lysis buffer (without inhibitors) and proteins eluted by the addition of Laemmli sample-reducing buffer. Proteins were separated by SDS-PAGE on a 4–15% gradient gel (BioRad) and transferred to a PVDF membrane (Millipore). Membranes were probed with Flag (Cat# 2368), MET pY1234/5 (Cat# 3077) (Cell Signaling Technologies).

Human MET knockout HeLa cells were transiently transfected with Lipofectamine3000 (Invitrogen) according to the manufacturer's protocols in a six-well plate format. Post transfection (24 hr), cells were washed with PBS (Gibco) (3 x washes) to remove serum and transfection media, and replaced with DMEM (Gibco) in the absence of any additives. Cells were serum starved for 4 hr, then stimulated with 50 ng/ml HGF (PeproTech) for 15 min at 37 °C, then immediately washed with cold PBS (3 x washes), and maintained on ice. Cells were then lysed in buffer (50 mM Tris-HCl, pH 7.5, 150 mM NaCl, 2 mM EDTA, and 1 % w/v Triton X-100 supplemented with protease inhibitor tablets (Roche), 1 mM sodium fluoride, and 1 mM sodium vanadate) on ice. Clarified whole cell lysates were run on an 8–16% SDS-PAGE gel (BioRad) and transferred to a nitrocellulose membrane (BioRad). Membranes were probed with MET pY1234/5 (Cat# 3077), Met (Cat# 8198), P-p44/42 MAPK Erk1/2 (T202/Y204) (Cat# 4376), p44/42 MAPK Erk1/2 (Cat# 4695), and β-Actin (Cat# 4970) (Cell Signaling Technologies).

## Ba/F3 proliferation assay

Ba/F3 cells stably expressing TPR-MET, TPR-METΔEx14, and empty MSCV constructs were seeded at 2.5E4 cells/ml in triplicate in a 94-well, round bottom plate for each time point in the presence and absence of 10 ng/ml IL-3. CellTiter-Glo reagent (Promega) was mixed at a 1:1 ratio with cells and luminescence was measured on a Veritas luminometer at 0, 48, and 96 hr post-seeding. In this study, we use a modified nomenclature, where we refer to TPR-MET as the TPR-fusion of MET with the full-length juxtamembrane sequence and TPR-METΔEx14 as the TPR-fusion lacking exon 14. Cell numbers were determined from a Ba/F3 cell and ATP standard curve generated according to the manufacturer's instructions. Data are presented as cell viability normalized to the fold change from the 0 hr time point.

For IL-3 titrations, Ba/F3 cells stably expressing TPR-MET, TPR-METΔEx14, and empty MSCV constructs were 3 x PBS washed and 5000 cells were seeded in a 94-well, round bottom plate. IL-3 was added to wells at 0–10 ng/ml (0, 0.078, 0.16, 0.31, 1.3, 2.5, 5, 10 ng/ml). CellTiter-Glo reagent

(Promega) was mixed at a 1:1 ratio with cells and luminescence was measured on a Veritas luminometer at 0, 24, and 48 hr after seeding and IL-3 addition.

Cell numbers for all proliferation assays were determined from a Ba/F3 and ATP CellTiter-Glo standard curve generated according to the manufacturer's instructions. Data are presented as cell viability normalized to the fold change from the 0 hr time point.

## MET kinase domain structural ensemble and RTK structural comparisons

Structural visualization, mapping, and analysis was completed using PyMOL unless otherwise stated. All human MET (UniProtKB accession: P08581) kinase domain crystal structures currently available were downloaded from the PDB. All PDB structures were loaded and globally aligned to generate the kinase domain ensemble. Residue distances were calculated from alpha-carbon x, y , and z coordinates and computationally analyzed. Raw PDB files were used to categorize structure features: resolution, construct boundaries, conformation, sequence features, mutations, and apo/holo states.

To choose representative active and inactive structures for score mapping and visualization, we generated an ensemble of 88 human MET kinase domain structures currently deposited in the PDB, and classified activity states based on alpha-carbon distances between catalytic site residues K1110, E1127, and F1223 (*Modi and Dunbrack, 2019*), with the majority of the MET KD structures in a 'BLBplus' or 'SRC-like' inactive conformation (*Modi and Dunbrack, 2019*). In the study, we refer to 3R7O, 3Q6W, and 4IWD as as our representative 'active' structures because they display classical active confirmation hallmarks αC-helix 'in,' K1110-E1127 salt bridge, DFG-in, solvent-exposed A-loop despite being inhibited. Within the ensemble, there is only one ATP-bound structure (PDB 3DKC), which harbors A-loop Y1234F and Y1235D stabilizing mutations, and also displays an inactive conformation. Within the group of inactive structures, there are two main conformational species based on DFG/αC-helix positioning and A-loop conformation: 'BLBplus' and 'BBAminus' (PDB 2G15 and PDB 5HTI represent the two species) (*Modi and Dunbrack, 2019*).

## RTK structural analysis and comparisons

Crystal structures of active and inactive human IR (PDB 4XLV, 4IBM), KIT (PDB 1 PKG, 1T45), EPHA3 (PDB 2QO9), RET (PDB 2IVT, 2IVS), MET (PDB 3R7O, 2G15), and AXL (PDB 5UAB) were obtained through the PDB. RMSD was calculated and plotted with the bio3D package in R for each kinase using the inactive structure as the reference.

For β5 positional comparison and Ramachandran analysis, PDB files were obtained for each kinase, and analyzed with the bio3D package in R to attain the Phi and Psi angles of each residue (*Grant et al., 2006*). The general and proline contour data was obtained from *Lovell et al., 2003* and plotted as an overlay with the specific kinase β5 residues aforementioned.

## Statistical filtering and classification of mutations

In this study, we classify gain-of-function (GOF) and loss-of-function (LOF) based on the following metrics.

First, the difference between the missense mutation score and the wild-type synonymous score must be smaller than the calculated propagated error of that mutation, within a selection condition, $\left(Score_{missense,IL3withdrawal} - Score_{\mu wt,IL3withdrawal}\right) \ll \sqrt{(SE_{missense,IL3\ withdrawal})^2 + (SE_{wt,IL3\ withdrawal})^2}$.

Second, missense mutations must be ≥±2 standard deviations (SD, σ) from the mean (μ) of wild-type synonymous mutation scores, $Score_{missense,IL3withdrawal} \geq 2\sigma$. Gain-of-function mutations are ≥+2 σand loss-of-function are ≥-2σ (*Figure 6—figure supplement 1A–B*).

To cross-compare mutations between IL-3 and IL-3 withdrawal conditions, as in *Figure 6—figure supplement 1D*, first the propagation of error between IL-3 and IL-3 withdrawal scores for the same variant was calculated $\sigma_x = \sqrt{(SE_{missense,IL3})^2 + (SE_{missense,IL3\ withdrawal})^2}$ , second the absolute difference between IL-3 and IL-3 withdrawal scores for the same variant was calculated $|\Delta Score| = |Score_{missense,IL3} - Score_{missense,IL3withdrawal}|$, then if the IL-3 withdrawal score was ≥+2 SDs, the IL-3 score was ≤0, and the absolute score difference between IL-3 and withdrawal conditions was larger than the propagated error, the mutational score was considered GOF, $Score_{missense,IL3withdrawal} \geq 2\sigma \cup Score_{missense,IL3} \leq 0 \cup |\Delta Score|\sigma_x$.

## Cancer and resistance mutation analysis

Cancer-associated missense mutations for the MET kinase domain was obtained from cBioPortal (NCBI ID: NM_000245). Resistance mutations were obtained from literature references (*Duplaquet et al., 2018*; *Fernandes et al., 2021*; *Saraon et al., 2021*; *Lu et al., 2017*; *Fujino et al., 2019*).

## Sequence alignments

All human RTK protein sequences used in alignments were acquired from UniProt (*Bateman et al., 2023*). Unless otherwise stated, alignments were done with MuscleWS using default parameters through JalView (*Waterhouse et al., 2009*), and amino acids were colored according to physicochemical properties, or percent sequence identity where noted.

Conservation scores were calculated through the Bio3D structural bioinformatics package in R (*Grant et al., 2006*) for each residue position by applying 'method = c('similarity')' and 'sub. matrix-=c('blosum62')' which compares amino acid similarity using a Blosum62 matrix.

## Acknowledgements

Sequencing was performed at the UCSF CAT, supported by UCSF PBBR, RRP IMIA, and NIH 1S10OD028511-01 grants. This work was supported by NIH CA239604 to EAC, NJ, JSF; HHMI Hanna Gray Fellowship and UCSF QBI Fellow program to WCM; and the UCSF Program for Breakthrough Biomedical Research, funded in part by the Sandler Foundation, to JSF.

## Additional information

### Competing interests

Natalia Jura: E.A.C. is a consultant at IHP Therapeutics, Valar Labs, Tatara Therapeutics and Pear Diagnostics, reports receiving commercial research grants from Pfizer, and has stock ownership in Tatara Therapeutics, HDT Bio, Clara Health, Aqtual, and Guardant Health. James S Fraser: N.J. is a founder of Rezo Therapeutics and a shareholder of Rezo Therapeutics, Sudo Therapeutics, and Type6 Therapeutics. N.J. is a SAB member of Sudo Therapeutics, Type6 Therapeutic and NIBR Oncology. The Jura laboratory has received sponsored research support from Genentech, Rezo Therapeutics and Type6 Therapeutics. The other authors declare that no competing interests exist.

### Funding

| Funder | Grant reference number | Author |
|---|---|---|
| National Cancer Institute | CA239604 | Eric A Collisson<br>Natalia Jura<br>James S Fraser |
| Howard Hughes Medical Institute | Hanna Gray Fellowship | Willow Coyote-Maestas |

The funders had no role in study design, data collection and interpretation, or the decision to submit the work for publication.

### Author contributions

Gabriella O Estevam, Conceptualization, Data curation, Formal analysis, Investigation, Methodology, Visualization, Writing – original draft, Project administration; Edmond M Linossi, Data curation, Formal analysis, Investigation, Methodology, Conceptualization, Writing – original draft, Project administration; Christian B Macdonald, Jennifer M Michaud, Formal analysis, Investigation, Conceptualization, Writing – original draft; Carla A Espinoza, Formal analysis, Investigation, Methodology, Conceptualization, Writing – original draft; Willow Coyote-Maestas, Data curation, Project administration, Writing – original draft; Eric A Collisson, Data curation, Funding acquisition, Project administration, Writing – original draft; Natalia Jura, Data curation, Funding acquisition, Writing – original draft, Project administration, Visualization; James S Fraser, Data curation, Supervision, Funding acquisition, Writing – original draft, Project administration, Visualization

### Author ORCIDs
Gabriella O Estevam http://orcid.org/0000-0002-9142-7805
Edmond M Linossi https://orcid.org/0000-0002-8039-573X
Christian B Macdonald https://orcid.org/0000-0002-0201-8832
Willow Coyote-Maestas https://orcid.org/0000-0001-9614-5340
Eric A Collisson https://orcid.org/0000-0001-8037-9388
Natalia Jura https://orcid.org/0000-0001-5129-641X
James S Fraser https://orcid.org/0000-0002-5080-2859

Reviewer #2 (Public Review): https://doi.org/10.7554/eLife.91619.3.sa1
Author response https://doi.org/10.7554/eLife.91619.3.sa2

## Additional files

### Supplementary files
• MDAR checklist

### Data availability
The sequencing data has been deposited at the NCBI SRA (PRJNA993160). Original data files and analysis source code is available at https://github.com/fraser-lab/MET_KinaseDomain_DMS (copy archived at *fraser-lab, 2023*).

The following dataset was generated:

| Author(s) | Year | Dataset title | Dataset URL | Database and Identifier |
|---|---|---|---|---|
| Estevam | 2023 | PRJNA993160 | https://www.ncbi.nlm.nih.gov/sra/?term=PRJNA993160 | NCBI Sequence Read Archive, PRJNA993160 |

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
