## [Editor Report · eLife assessment]

This manuscript describes a deep mutational scanning study of the kinase domain of the MET receptor tyrosine kinase. The study yields an **important** catalog of essentially all possible deleterious mutations in this portion of the receptor., with **convincing** evidence. The manuscript will be of interest to researchers working in the field of receptor tyrosine kinases.

---

## [Referee Report · Reviewer #2 (Public Review)]

Summary:

The authors describe a deep mutational scanning (DMS) study of the kinase domain of the c-MET receptor tyrosine kinase. The screen is conducted with a highly activated fusion oncoprotein - Tpr-MET - in which the MET kinase domain is fused to the Tpr dimerization element. The mutagenized region includes the entire kinase domain and an alpha-helix in the juxtamembrane region that is essentially part of the MET kinase domain. The DMS screen is carried out in two contexts, one containing the entire cytoplasmic region of MET, and the other with an "exon 14 deletion" which removes a large portion of the juxtamembrane region (but retains the aforementioned alpha-helix). The work provides a robust and essentially exhaustive catalog of the effect of mutations (within the kinase domain) on the ability the Tpr-MET fusion oncoproteins to drive IL3-independent growth of Ba/F3 cells. Every residue in the kinase is mutated to every natural amino acid. Given the design of the screen, one would expect it to be a powerful tool for identifying mutations that impair catalytic activity and therefore impair IL3-independent proliferation. This is borne out by the data, which reveal many many deleterious mutations. The study reveals relatively few "gain-of-fitness" mutations, but this is not unexpected because it is carried out with an already-activated form of the MET kinase (the oncogenic Tpr-met fusion).

Strengths:

The authors take a very scholarly and thorough approach in interpreting the effect of mutations in light of available information for the structure and regulation of MET and other kinases. They examine the effect of mutations in the so-called catalytic (C) and regulatory (R) spines, the interface between the JM alpha-helix and the C-helix, the glycine-rich loop and other key elements of the kinase, providing a structural rationale for the deleterious effect of mutations. Comparison of the panoply of deleterious mutations in the TPR-met versus TPR- exon14del-MET DMS screens reveals an interesting difference - the exon14 deletion MET is much more tolerant of mutations in the JM alpha-helix/C-helix interface. The reason for this is unclear, however.

An important qualification of the study is that it was carried out with the already highly activated Tpr-Met fusion. As a consequence, it is not expected to reveal mutations that activate the kinase -- activate in the sense of promoting a switch between physiologically-relevant inactive and active states. Consistent with this, the authors note that gain-of-fitness mutations are rare in their screen, and those that are identified induce modest but significant increases in fitness.

---

## [Author Response]

The following is the authors’ response to the original reviews.

**Public Reviews:**

**Reviewer #1 (Public Review):**
Summary:This manuscript by Estevam et al. reports new insights into the regulation of the receptor tyrosine kinase MET gained from two deep mutational scanning (DMS) datasets. In this paper, the authors use a classic selection system for oncogenic kinase signaling, the murine Ba/F3 cell line, to assess the functional effects of thousands of mutations in the kinase domains of MET in two contexts: (1) fusion of the whole MET intracellular region to the dimerization domain TPR, and (2) the same fusion protein, but with exon 14, which encodes part of the juxtamembrane region of MET, skipped. Critically, exon 14 skipping yields a version of MET that is found in many cancers and has higher signaling activity than the canonical MET isoform. The authors extensively analyze their DMS data to very convincingly show that their selection assay reports on kinase activity, by illustrating that many functionally important structural components of the kinase domain are not tolerant of mutation. Then, they turn their attention to a helical region of the juxtamembrane region (αJM), immediately after exon 14, which is posited to play a regulatory role in MET. Their DMS data illustrate that the strength and mutational tolerance of interactions between αJM and the key αC helix in the kinase domain depends on the presence or absence of exon 14. They also identify residues in the N-lobe of the kinase, such as P1153, which are not conserved across tyrosine kinases but appear to be essential for MET and MET-like kinases. Finally, the authors analyze their DMS data in the context of clinically-observed mutations and drug-resistance mutations.Overall, this manuscript is exciting because it provides new insights into MET regulation in general, as well as the role of exon 14. It also reveals ways in which the JM region of MET is different from that of many other receptor tyrosinekinases. The exon 14-skipped fusion protein DMS data is somewhat underexplored and could be discussed in greater detail, which would elevate excitement about the work. Furthermore, some of the cell biological validation experiments and the juxtaposition with clinical data are perhaps not assessed/interpreted as clearly they could be. Some constructive suggestions are given below to enhance the impact of the manuscript.Strengths:The main strengths of this paper, also summarized above in the summary, are as follows:(1) The authors very convincingly show that Ba/F3 cells can be coupled with deep mutational scanning to examine MET mutational effects. This is most clearly shown by highlighting how all of the known kinase structure and regulatory elements are highly sensitive to mutations, in accordance with a few other DMS datasets on other kinases.(2) A highlight of this paper is the juxtaposition of two DMS datasets for two different isoforms of the MET receptor. Very few comparisons like this exist in the literature, and they show how small changes to the overall architecture of a protein can impact its regulation and mutational sensitivity.(3) Another exciting advance in this manuscript is the deep structural analysis of the MET juxtamembrane region with respect to that of other tyrosine kinases - guided by the striking effect of mutations in the juxtamembrane helical region. The authors illustrate how the JM region of MET differs from that of other tyrosine kinases.(4) Overall, this manuscript will provide a resource for interpreting clinically relevant MET mutations.Weaknesses:(1) The manuscript is front-loaded with extensive analysis of the first DMS dataset, in which exon 14 is present, however, the discussion and analysis of the exon 14-skipped dataset is somewhat limited. In particular, a deeper discussion of the differences between the two datasets is warranted, to lay out the full landscape of mutations that have different functional consequences in the two isoforms. Rather, the authors only focus on differences in the JM region. What are the broader structural effects of exon 14 skipping across the whole kinase domain?

Thank you for your feedback on our manuscript and our analysis of the exon 14 skipped mutational scanning data. The lack of a robust growth differential between the wild type MET intracellular domain and the exon 14 skipped isoform within the Ba/F3 system suggests that there is not a significant growth advantage related to exon 14 skipping, likely due to the constitutive activation of both constructs by the TPR domain, which also suggests that the assay is potentially less sensitive to nuanced JM-driven effects between these two isoforms, aside from the highly sensitive ⍺JM-helix. We also lose insight on membrane-related interactions imposed on the juxtamembrane that may be important to fully understand the differences between these two isoforms in the cytoplasmically-expressed context. Therefore, we can at most speculate exon 14 skipped related differences between these two datasets.

With these caveats in mind, to further address exon 14 and juxtamembrane-driven differences between these two mutational landscapes, we calculated the absolute score difference between TPR-METΔEx14 and TPR-MET (|METΔEx14 - MET|) and plotted the |ΔScore| in a heatmap. Overall, the two landscapes, as noted in the text, are largely similar with differences emerging mostly for specific mutations. Where we see the largest secondary structural difference continues to be the ⍺JM-helix, where MET is more sensitive to helix-breaking mutations such as proline. Again L1062 has the greatest difference in sensitivity between these two datasets for the ⍺JM-helix, with the introduction of negative charge resulting in loss-of-function for the TPR-MET kinase domain but having a null effect in the TPR-METΔEx14 kinase domain. Other positions with strong differences include the ⍺G and APE motif.

We have incorporated more detailed discussion in text.

(2) It is unclear if gain-of-function mutations can actually be detected robustly in this specific system. This isn't a problem at face value, as different selection assays have different dynamic ranges. However, the authors don't discuss the statistical significance and reproducibility of gain- vs loss-of-function mutations, and none of the gain-of-function mutations are experimentally validated (some appear to show loss-of-function in their cellular validation assay with full-length MET). The manuscript would benefit from deeper statistical analysis (and discussion in the text) of gain-of-function mutations, as well as further validation of a broad range of activity scores in a functional assay. For the latter point, one option would be to express individual clones from their library in Ba/F3 cells and blot for MET activation loop phosphorylation (which is probably a reasonable proxy for activity/activation).

Thank you for your comment on the statistical interpretations of gain-of-function (GOF) and loss-of-function (LOF) mutations. In this study we classify GOF and LOF based on the following metrics:

(1) The difference between the missense mutation score and the wild type synonymous score for a given position must be smaller than the calculated propagated error, for both IL-3 withdrawal and IL-3 conditions

(2) Missense mutations must be ≥ ±2 standard deviations (SD) from the mean of wild type synonymous mutations

Given that our assay was conducted in a constitutively active kinase in the TPR-fusion context, gain-of-function mutations are expected to not only be rare, but also supersede baseline fitness. Within the IL-3 conditions, we expect that cells are not reliant or “addicted” to MET for growth proliferation. Nevertheless, due to the parallel nature of the screen, we can compare scores for variants in the IL-3 control and IL-3 withdrawal conditions to filter mutations that are solely exhibiting high fitness under selective pressure.

To identify these mutations we (1) calculated the propagation of error between IL-3 and IL-3 withdrawal scores for the same variant (2) calculated the absolute difference between IL-3 and IL-3 withdrawal scores for the same variant (3) filtered variants if the IL-3 withdrawal score was ≥ +2 SDs, the IL-3 score was ≤ 0, and the absolute score difference between IL-3 and withdrawal conditions was larger than the propagated error.

In analyzing mutations within the IL-3 withdrawal conditions, applying our statistical metrics, we find 33 mutations within the MET library, and 30 in the METΔEx14 library, that have a score of ≥ +2 SD and low propagated error. By increasing our boundary to ≥+2.5 SD, we can classify mutations with even higher confidence, identifying 10 mutations within the MET library, and 9 in the METΔEx14 library (Supplemental Data Figure 7).

(3) In light of point 2, above, much of the discussion about clinically-relevant gain-of-function mutations feels a bit stretched - although this section is definitely very interesting in premise. A clearer delineation of gain-of-function, with further statistical support and ideally also some validation, would greatly strengthen the claims in this section.

To address this concern, we have provided additional analysis and details on gain-of-function (GOF) classification in Supplemental Data Figure 5 and the overlap between GOF and clinically associated mutations in Supplemental Data Figure 8. Within our gain-of-function classifications, we pick up on several mutations at positions that have been clinically detected and experimentally validated in previous studies in both libraries (D1228, G1163, L1195), and show that GOF mutations also have low variance.

**Reviewer #2 (Public Review):**
Summary:The authors describe a deep mutational scanning (DMS) study of the kinase domain of the c-MET receptor tyrosine kinase. The screen is conducted with a highly activated fusion oncoprotein - Tpr-MET - in which the MET kinase domain is fused to the Tpr dimerization element. The mutagenized region includes the entire kinase domain and an alpha-helix in the juxtamembrane region that is essentially part of the MET kinase domain. The DMS screen is carried out in two contexts, one containing the entire cytoplasmic region of MET, and the other with an "exon 14 deletion" which removes a large portion of the juxtamembrane region (but retains the aforementioned alpha-helix). The work provides a robust and essentially exhaustive catalog of the effect of mutations (within the kinase domain) on the ability of the Tpr-MET fusion oncoproteins to drive IL3-independent growth of Ba/F3 cells. Every residue in the kinase is mutated to every natural amino acid. Given the design of the screen, one would expect it to be a powerful tool for identifying mutations that impair catalytic activity and therefore impair IL3-independent proliferation, but not the right tool for identifying gain-of-function mutations that operate by shifting the kinase from an inactive to active state (because the Tpr-Met fusion construct is already very highly activated). This is borne out by the data, which reveal many many deleterious mutations and few "gain-of-function" mutations (which are of uncertain significance, as discussed below).Strengths:The authors take a very scholarly and thorough approach to interpreting the effect of mutations in light of available information for the structure and regulation of MET and other kinases. They examine the effect of mutations in the so-called catalytic (C) and regulatory (R) spines, the interface between the JM alpha-helix and the C-helix, the glycine-rich loop, and other key elements of the kinase, providing a structural rationale for the deleterious effect of mutations. Comparison of the panoply of deleterious mutations in the TPR-met versus TPR- exon14del-MET DMS screens reveals an interesting difference - the exon14 deletion MET is much more tolerant of mutations in the JM alpha-helix/C-helix interface. The reason for this is unclear, however.Weaknesses:Because the screens were conducted with highly active Tpr-MET fusions, they have limited power to reveal gain-of-function mutations. Indeed, to the extent that Tpr-MET is as active or even more active than ligand-activated WT MET, one could argue that it is "fully" activated and that any additional gain of fitness would be "super-physiologic". I would expect such mutations to be rare (assuming that they could be detected at all in the Ba/F3 proliferation assay). Consistent with this, the authors note that gain-of-function mutations are rare in their screen (as judged by being more fit than the average of synonymous mutations). In their discussion of cancer-associated mutations, they highlight several "strong GOF variants in the DMS". It is unclear what the authors mean by "strong GOF", indeed it is unclear to this reviewer whether the screen has revealed any true gain of function mutations at all. A few points in this regard:(1) More active than the average of synonymous mutations (nucleotide changes that have no effect on the sequence of the expressed protein) seems to be an awfully low bar for GOF - by that measure, several synonymous mutations would presumably be classified as GOF.

We completely agree that any mutation above the average synonymous would not be a robust assessment and thus why we statically filtered mutations in our entire analysis. To this point, and that of Reviewer 1, we have further outlined our statistical definitions. In classifying mutations as GOF or LOF, the following parameters were used:

(1) The difference between the missense mutation score and the wild type synonymous score for a given position must be smaller than the calculated propagated error, for both IL-3 withdrawal and IL-3 conditions

(2) Missense mutations must be ≥ ±2 standard deviations (SD) from the mean of wild type synonymous mutations

Therefore, only variants at the tail-ends of the mutational distribution were assessed, and further filtered based on propagation of error. For this reason, a “strong GOF” mutation as noted in this study is one that improves the fitness of an already active kinase. As pointed out, within our analysis, these are very rare occurrences, and in focusing on cancer-associated mutations we find that the variants that meet these statistical parameters require a larger genetic “leap” in the codon space. Overall, we have also changed our language in reference to GOF mutations in text.

We hope this concern has been addressed in the new Supplemental Data Figures.

(2) In the +IL3 heatmap in supplemental Figure 1A, there is as much or more "blue" indicating GOF as in the -IL3 heatmap, which could suggest that the observed level of gain in fitness is noise, not signal.

We hope this concern has been addressed in the previous responses and new Supplemental Data Figures.

(3) And finally, consistent with this interpretation, in Supplemental Figure 1C, comparing the synonymous and missense panels in the IL3 withdrawal condition suggests that the most active missense mutations (characterized here as strong GOF) are no more active than the most active synonymous mutations.

We hope this concern has been addressed in the previous responses and figures above.

My other major concern with the work as presented is that the authors conflate "activity" and "activation" in discussing the effects of mutations. "Activation" implies a role in regulation - affecting a switch between inactive and active conformations or states - at least in this reviewer's mind. As discussed above, the screen per se does not probe activation, only activity. To the extent that the residues discussed are important for activation/regulation of the kinase, that information is coming from prior structural/functional studies of MET and other kinases, not from the DMS screen conducted here. Of course, it is appropriate and interesting for the authors to consider residues that are known to form important structural/regulatory elements, but they should be careful with the use of activity vs. activation and make it clear to the reader that the screen probes the former. One example - in the abstract, the authors rightly note that their approach has revealed a critical hydrophobic interaction between the JM segment and the C-helix, but then they go on to assert that this points to differences in the regulation of MET and other RTKs. There is no evidence that this is a regulatory interaction, as opposed to simply a structural element present in MET and indeed the authors' examination of prior crystal structures shows that the interaction is present in both active and inactive states.

Thank you, and we completely agree that the distinction between “activity” and “activation” is important and that we can at most speculate and propose models for effects related to activation from this screen. We have edited the text to reflect these distinctions. In respect to activation and the second point, we believe the screen highlights the ⍺JM-C interface as a critical structural region, which *may* have a role in regulation based on the paradigm of juxtamembrane regulation in RTKs, the presence of a similar interface in TAM family kinases, the co-movement of the ⍺JM-helix and ⍺C-helix between active and inactive conformations in the structural ensemble, and the observation that within the TPR-METΔEx14 library there is a greater tolerance for mutations at interface positions than TPR-MET. We hope that are follow-up studies that directly probe the ⍺JM-C interface in respect to the entire juxtamembrane to truly say if/ what role this conserved motif plays in regard to MET function. We have changed the language of the text to reflect how these differences contribute to our proposed model, rather than any unintended assertion on direct regulatory effects.

**Recommendations for the authors:**

**Reviewer #1 (Recommendations For The Authors):**
Suggested major points to address:(1) Although the authors show that several key functional residues in the kinase domain are highly sensitive to mutation, it would be nice if the authors further established a clear connection between kinase activity and enrichment in the Ba/F3 assay. Specifically, it is unclear to what extent there is a correlation between the extent of enrichment/depletion and kinase activity - is a larger activity score necessarily indicative of higher kinase activity? This is partly validated by the P1153L mutation autophosphorylation western blots in Figure 4B, but this correlation is somewhat undermined by the data in 5F. Autophosphorylation data (or phosphorylation data on a direct downstream substrate) for a few mutants would really solidify what the activity score is truly reporting. This might also clarify the extent to which the difference between the two screens can be interpreted, and the extent to which gain-of-function can be interpreted.

The Ba/F3 assay was carefully chosen for its addiction to exogenous IL-3, which serves as a permissive signaling switch. Any mutation that prevents TPR-MET/ΔEx14 from properly functioning is therefore dampening its signaling ability. Nevertheless, it is possible that some mutations with high scores are truly improving activity and others are sustaining activity through more stable interactions than the wild type kinase domain or with downstream signaling partners, which would require careful biochemical dissection outside the scope of this study. To address these points, we now refer to the mutation score simply as “score” rather than “activity score” and further discuss these caveats in text.

(2) Overall, the exon 14-skipped dataset is under-discussed in the paper. The comparison of the two datasets is where most deep insights are likely to be found, and so a more thorough analysis/discussion of this dataset would really elevate the significance of the paper. For example, there appear to be a very large number of mutations that have divergent effects in the two screens (everything along the dashed lines in Figure 5D), but it's unclear where most of these mutations lie on the structure. It would be helpful if the residues with divergent mutational effects between the two screens (Supplementary Figure 5E) were mapped onto a structure of the JM-KD construct.

To address this concern, new analysis has been added to the supplement, showing the score differences between MET and METΔEx14 mutations as a heatmap (Supplemental Data Figure 7A). Within this analysis we further applied our statistical filtering methods and structurally mapped positions with the greatest differential scores to show where divergent effects cluster (Supplemental Data Figure 7D). Consistent with our previous reports, the ⍺JM-helix and ⍺C-helix show the largest cluster of divergent effects, in addition to sites such as the ⍺G and APE motif. Further discussion of these points have been added to the text.

(3) Based on the observations that αJM-αC interactions seem to be less strictly required in the exon 14 mutant, the hypothesis that exon 14 skipping merely removes a Cbl docking site seems largely unsatisfactory. There seems to be more direct structural alterations that could explain this change, but these are not really discussed or speculated on. Related to this, while L1062 mutations are more tolerated, as the authors showed in both the mutational heatmap and the cellular experiments, its binding counterpart L1125 still seems to be somewhat immutable based on the heatmaps. So, more hypothesis/exploration of how exon 14 skipping affects MET KD structure would be a nice addition to the paper.

We agree that loss of the Cbl docking site is an insufficient model to capture the full nature of JM regulation and exon 14 skipping effects, which was a major incentive for this study. The outstanding ⍺JM-⍺C-helix sensitivity also excites us because it points to a potential regions of the JM that potentially is involved in kinase activity through ⍺C-helix interactions, much like the CDK models and other RTK-JM interactions. We observed that the ⍺JM-⍺C helix retain contact, and propose that the ⍺JM-⍺C helix move in unison between active and inactive conformations. However, it is possible that a more complicated mechanism might also exist, where there is a larger degree of maintenance of these contacts in a homodimer. For instance, in Figure 3G, if you compare the ⍺JM-helix conformations, in both RON and AXL there is more distance and a pivot away from the ⍺C-helix. It’s is possible that there are shared mechanisms between the MET and TAM families that could further elucidate exactly how these ⍺JM-helices interact with the kinase domain during the activity transitions and what biophysical role JM truncations play.

(4) The discussion about mutations S1122Q and L1062D is a bit confusing and incomplete. From the DMS data, it appears that L1062D should be mildly gain-of-function for the exon 14 deletion variant and very loss of function for wild-type MET. In the validation HeLa cell experiments L1062D is loss-of-function in both contexts, but a mention of this discrepancy is omitted. Then, when the discordance between DMS and HeLa cell experiments is observed again for S1122Q, it is explicitly called out for activation-loop phosphorylation, but then there is no mention of the fact that HGF stimulation leads to greater pERK levels for S1122Q in the exon 14 deletion context (the opposite of the DMS result). The Erk phosphorylation discrepancy should be mentioned. It is entirely reasonable, as the authors suggest, that there are differences between full-length MET and the TPR fusions, but the enhanced Erk phosphorylation by the S1122Q mutation is surprising (and intriguing!). This section could use some re-analysis/re-writing and further discussion.

Thank you for this comment. As noted L1062D shows slight GOF in METΔEx14 but LOF in MET. The blots show expression of L1062D and S1122Q in the full length receptor in the absence and presence of HGF stimulation. L1062D is loss of function for both contexts only in -HGF conditions, but shows expression in phosphorylated METΔEx14, but not MET. For S1122Q, indeed there is a stronger pERK signal in the METΔEx14, which highlights how probing all regions of phosphorylation (A-loop and C-tail) and many MET-associate pathways (ERK, AKT) may be important to understand in what way these mutations are affect MET phosphorylation and proliferation. We have included this point in the text.

(5) Related to the previous point, one other thing to consider here is that perhaps gain-of-function mutations are simply not detectable in this particular DMS assay. The authors state that GOF and LOF are defined as 2 standard deviations from the mean of the WT-synonymous distribution. How many mutations are actually designated to be GOF based on this criterion? Are those GOF mutations as reproducible as the LOF mutations? It would be worthwhile to separately analyze the variance in activity scores for every loss-of-function mutation and gain-of-function mutation. It seems likely that loss-of-function scores are a lot more reproducible than gain-of-function ones, suggesting that the most apparent gain-of-function signal is just noise in the assay. The few outliers to this point (true gain-of-function mutations) may be some of the ones discussed in Figure 6. If this is true, it would lend confidence to the claims associated with Figure 6.

In analyzing and classifying both GOF and LOF mutations, error was a main filtering parameter. Each fitness score, calculated by Enrich2, is representative of the slope across time points and biological replicates for the read frequency of the mutation. The associated standard error (SE) reflects the variance for each mutation within the scoring framework (Rubin et al., 2017). Mutations were then further filtered based on low propagated error, calculated by comparing the standard error (SE) of each missense mutation to the SE of the respective wild type synonymous mutation. Therefore, mutations were only classified as GOF or LOF if there was low error, in addition to the other score filters previously described. We have plotted the classified GOF mutations with their respective SE in the newly incorporated Supplemental Data Figure 8C.

(6) In the discussion of panels 6C and 6D, the assertion is that the "clinical, not validated" category has more mutations that are low-fitness outliers than the "clinical, validated" category. From the graphs, it's actually hard to tell if this is the case for two reasons: (1) the way the graphs are normalized, (to the largest value in each histogram), you cannot compare bar heights (and thus number of mutations) between two histograms on the same graph. (2) Just looking at the shapes of the distributions, or considering maybe the mean or median values, it's unclear whether the "validated" and "not validated" populations are actually different from one another.

This is an important indication, and we have added analysis showing the distribution and number of clinically-associated mutations within our libraries without normalization in the main text and in Supplemental Data Figure 8A-B.

(7) This sentence in the last results section is somewhat unclear: "GOF resistance mutations may indicate an effect on the equilibrium of kinase activation, whereas LOF resistance mutations likely affect inhibitor-protein interactions directly." The first part makes sense, but it is not totally obvious how one can infer anything about inhibitor-protein interactions from mutations that are LOF with respect to kinase activity. Related to this, how are LOF mutations selected in the presence of an inhibitor? Is the assumption here that the mutation might totally abrogate inhibitor binding but only slightly impair the kinase? Perhaps this could be explained a bit more.

Here, the idea we wanted to get across is that there are two models that can explain how a mutation can contribute to resistance: shift the activity equilibrium at baseline or directly impair drug effects and restore baseline activity. Mutations that are labeled resistant and GOF, favor the first model. Mutations that are labeled resistant and LOF, favor the second model. In the presence of an inhibitor, which is in the scope outside of this study, LOF mutations would be sensitive to the inhibitor (ie WT-like and sensitive).

(8) Some additional details of the library preparation and sequencing should be given in the methods section. It appears that the variable region of the library is roughly 275 amino acid residues long, which means >800 bases. How was this sequenced? From the methods, it sounds like all of the variants were pooled into a single library, but then sequencing was done using a 300x300 paired-end Illumina kit, which would not cover the length of the whole variable region. Was the library actually screened in segments as sub-libraries and then separately sequenced? Alternatively, was the whole library screened at once, and then different segments were amplified out for sequencing? If the latter approach is used, this could yield confounding results for counting wild-type variants that have the parent wild-type coding sequence. For example, if you amplify your kinase library in three segments after a single selection on the whole library, and you sequence those three segments separately, you might find a read that appears as wild-type in the part you amplified/sequenced but has a mutation in a region that you did not sequence. If this approach is taken, the counts for the wild-type sequence would be inaccurate, in which case, how is the data normalized with WT as a reference? Regardless of the method used, some more details should be provided in the methods section.

In this study, we used the Nextera XT DNA Library Preparation Kit (Illumina), which uses a tagmentanation approach that randomly fragments our 861 bp amplicon into ~300 bp fragments with a transposase, resulting in a Poisson distribution of fragment sizes. This allows for direct sequencing of all amplicons and libraries with an SP300 paired-end run, which we ran on two lanes of a NovaSeq6000. Samples are demultiplexed and processed by our analysis pipeline with a lookup table that associates the unique dual index to the specific sample (library, time point, biological replicate, IL-3 condition).

The TPR-MET and TPR-METΔEx14 libraries were prepared in parallel throughout the entire experiment, from cloning to virus generation to transductions, screening, cell harvesting, sequencing prep, and sequencing. In other words, the TPR-MET and TPR-METΔEx14 were transduced into their own, respective batch of cells for each biological replicate, then selected and screened on the same day for each replicate and time point. Each library and condition (time point, biological replicate, IL-3 condition) was prepared in parallel but still an independent sample. At the stage of tagmentation, each sample was arrayed, where each well corresponds to a library, biological replicate, and time point. At the stage of sequencing, samples across the two libraries were normalized to 10mM (library, biological replicate, time point, IL-3 condition) then pooled together and all run on two lanes of the same NovaSeq6000 flow cell.

PCR and sequencing bias was one of the most important parameters for us, which is why we performed tagmentation in parallel and sequenced everything on the same run. We have added extra details to the methods and hope that we have clarified your questions on this matter.

Suggested minor points to address:(1) TPR (as in TPR-MET fusion) is not defined in the text when it is first mentioned. And it wasn't immediately clear that this is not a membrane-associated domain (Figure 5E makes this way more obvious than Figure 1B does). Perhaps this could be made more explicit in the text or in Figure 1.

We have incorporated a new schematic in Figure 1B to better illustrate the TPR-fusion constructs used within this study. The usage of the TPR-fusion is first mentioned in the introduction, paragraph 4, and revised the main-text to delineate the usage of the TPR-fusion more clearly.

(2) In Figure 2G, it would be helpful if the wild-type amino acid residue was listed underneath the position number in the two graphs (even though those residues are also highlighted in 2H).

Thank you for this recommendation, we have added the wild type amino acid next to the position number in the x-axis label.

(3) For Supplementary Data Figure 2, is it possible to calculate conservation scores at each position using some kind of evolutionary model, rather than relying on visual inspection of the sequence logo? Can one quantitatively assert that the C-spine is less conserved than the R-spine overall, or can this only be said for certain positions? Related to this, in comparing Figure 2G to Supplementary Data Figure 2, it is interesting that there isn't any obvious correspondence between mutational tolerance and conservation within the C-spine. For example, 1165 seems to be the most conserved position in the C-spine, but several substitutions are tolerated at this position, just like 1210, which is one of the least conserved positions in the C-spine. Finally, it's very likely that positions 1165, 1210, 1272, and 1276 co-vary, given that they all pack into the same hydrophobic cluster. This might be why they appear less conserved. These last few points might be worth discussing briefly if the authors want to relate mutational tolerance to evolutionary conservation.

Thank you for this recommendation. To better quantitatively determine C-spine versus R-spine conservation, we performed a multiple sequence alignment of all RTK kinase domain sequences to properly identify corresponding R- and C-spine locations, as previously done in generating the spine logos, then used the bio3D structural bioinformatics package in R to calculate the conservation score of each residue position by amino acid “similarity” with a blosum62 matrix (Supplemental Data Figure 2B). In concordance with the logos, we find that C-spine positions 1092, 1108, 1165 have the highest conservation scores, even compared to some R-spine mutations. We also see across the alignment that indeed, C-spine positions 1165 1210,1211,1212, and 1272, and 1276 co-vary within RTK families. We have revised the text to reflect these points, and more specifically discuss position-level conservation rather than generalizing conservation for the C- and R-spines.

(4) On Page 7 of the merged document, there appear to be some figure labeling errors. In the first and second paragraphs of the "Critical contacts between..." section, Figure 3B is referenced multiple times as a structural alignment/ensemble, but this is a heatmap.

Thank you for catching this! The correct figure panels are now referenced.

(5) In the text describing Figure 3A, it is stated that the structures were aligned to the N-lobe, but the figure legend says that all structures were aligned to alpha-C and alpha-JM.

Thank you - a local alignment to the ⍺JM-helix and ⍺C-helix is correct, the idea here being that if the ⍺JM-helix and ⍺C-helix are linked to an active/inactive conformation like in the case of the insulin receptor, these two clusters could be revealed through the structural ensemble. However, we discovered this was not the case, combined with the DMS sensitivity to mutations at the packing interface leads us to believe that the MET JM has a distinctive regulatory mechanism that relies on this ⍺C-helix interface. We have made this correction to the text.

(6) It would be helpful if the alpha-C and alpha-JM helices in Figure 3D were labeled on the MET structures.

The ⍺C-helix and ⍺JM-helix are now labeled in Figure 3D.

(7) It appears that Figure 4E is never explicitly referenced in the text.

Thank you, Figure 4E is now appropriately referenced in the text.

(8) Throughout the Figure 6 legend, for the histograms, it is stated that "Counts are normalized to the total mutations in each screen dataset." This might not be the correct description of normalization, as this would mean that the sum of all of the bins should equal 1. Rather, the normalization appears to be to the bin with the largest number of mutants in it, which is given a value of 1. This difference is really critical to how one visually inspects the overlaid histograms.

Thank you for this comment. Here, the intention was to aid in the visualization of the distribution of cancer-associated and resistance associated mutations, which is a much smaller population compared to the whole library and becomes easily masked. We originally applied a “stat(ncount)” function in R, which as noted scales the data and sets the peak to 1, which only applied to the clinical and cancer-associated mutations plotted. Now, to better compare distributions, normalization has been removed, instead opting to overlay the distributions of all missense mutations and the subset of clinical mutations directly with their own y-axis scale. This modification has been made throughout Figure 6 panels, hopefully improving interpretability.

**Reviewer #2 (Recommendations For The Authors):**
A few thoughts/suggestions:(1) Regarding kinase regulation, the "closing of N- and C-lobe" upon activation is an often mentioned component of activation, and I'm sure is true in many cases, but it is not a general feature of kinase activation.

The text has been updated - we removed the description of N- and C-lobe closure.

(2) With respect to the inactive state of MEK, the DFG-flipped structure discussed here is almost certainly an inhibitor-induced conformation. Again, DFG-flip is often discussed as a mechanism of kinase regulation, and while in some kinases this might be the case, more often it is a drug-induced or drug-stabilized inactive conformation. The SRC/CDK-like inactive conformation in 2G15 is more likely a physiologically relevant inactive state. (or even better, the ATP-bound inactive state structure 3DKC, which exhibits a somewhat different SRC/CDK-like inactive conformation).

The PDB 3R7O structure was chosen as the main representation because it was the clearest representation of a wild type structure with an aligned R- and C- spine, solvent-exposed, phosphorylated activation loop. Although 3DKC is bound to ATP, this structure is still in an inactive conformation and has stabilizing mutations (Y1234/F, Y1235D) and an atypical alpha helix structure in the activation loop. However, we agree the SRC/CDK-like inactive conformation is an important representation and we have incorporated our structural mapping on 2G15 in the new supplemental figures with further details on statistical analysis and comparison of libraries.

(3) Following the comments above, I would describe the process of activation in a simpler way (in any case, it is peripheral to the work described here). Something along the lines of "phosphorylation on tyrosines XX and XX induces rearrangement of the activation segment and promotes and stabilizes the inward active position of the C-helix." Can go on to mention that this forms the E1127/K1110 salt bridge. (The DFG is already "in" in the SRc/CDK-like inactive state).

We have changed the language to more simply describe activation. Thank you!

(4) Would be great to see DMS with the intact receptor done in a way that could identify mutations that lead to activation in a ligand-independent manner. (but obviously beyond the scope of this paper).

Agreed! This would be an excellent follow up for the future, especially to elucidate juxtamembrane regulation, as the membrane context is likely required.

A typo or two:Boarded instead of bordered/outlined in legend to Fig. 1.P11553L in the 2nd line of the 2nd paragraph in that section.

Thank you, we have addressed these typos!